# High-resolution CTCF footprinting reveals impact of chromatin state on cohesin extrusion

Corriene E. Sept[1,2,3], Y. Esther Tak[4,5], Viraat Goel [3,6,7,8], Mital S. Bhakta[9], Christian G. Cerda-Smith [10], Haley M. Hutchinson [10], Marco Blanchette[11], Christine E. Eyler[12,13], Sarah E. Johnstone[3,14], J. Keith Joung[4,5], Anders S. Hansen [3,6,7,8] & Martin J. Aryee [1,2,3,15] ✉

Cohesin-mediated DNA loop extrusion enables gene regulation by distal enhancers through the establishment of chromosome structure and long-range enhancer-promoter interactions. The best characterized cohesin-related structures, such as topologically associating domains (TADs) anchored at convergent CTCF binding sites, represent static conformations. Consequently, loop extrusion dynamics remain poorly understood. To better characterize static and dynamically extruding chromatin loop structures, we use MNase-based 3D genome assays to simultaneously determine CTCF and cohesin localization as well as the 3D contacts they mediate. Here we present CTCF Analyzer (with) Multinomial Estimation (*CAMEL*), a tool that identifies CTCF footprints at near base-pair resolution in CTCF MNase HiChIP. We also use Region Capture Micro-C to identify a CTCF-adjacent footprint that is attributed to cohesin occupancy. We leverage this substantial advance in resolution to determine that the fully extruded (CTCF-CTCF loop) state is rare genome-wide with locus-specific variation from ~1–10%. We further investigate the impact of chromatin state on loop extrusion dynamics and find that active regulatory elements impede cohesin extrusion. These findings support a model of topological regulation whereby the transient, partially extruded state facilitates enhancer-promoter contacts that can regulate transcription.

The three-dimensional organization of the genome within the nucleus is a key regulator of gene expression. The cohesin complex, through its loop extrusion ability that brings distant genomic regions into close physical proximity, is a key determinant of this spatial arrangement[1]. Cohesin-mediated loop extrusion is a modulator of transcriptional programs through its ability to both facilitate and block interactions between enhancers and promoters. As a consequence, disruptions to cohesin-mediated loop extrusion can lead to misregulation of gene expression and are implicated in various genetic diseases[2–5].

CCCTC-binding factor (CTCF) can act as an extrusion barrier through its ability to bind and stabilize cohesin on DNA[6], serving to preferentially localize and anchor one or both ends of cohesin loops. Through its interaction with cohesin, CTCF can act to promote or inhibit gene expression in a context dependent manner. For example, many topologically associated domains (TADs) are bounded by convergently oriented CTCF sites and insulate the genes contained within from activation by enhancers located outside the TAD[7–9]. On the other hand, promoter-proximal CTCF binding sites can enable gene activation by distal enhancers[10–12], although the mechanisms involved are not well understood.

While the best understood cohesin-related structures, including TADs and enhancer-promoter loops, represent static conformations,

loop extrusion has also been shown to be a highly dynamic process with an extrusion rate of ~1 kb/s[13]. Recent live cell-imaging studies of loop extrusion dynamics at two TADs bounded by convergent CTCF sites found that only a small fraction (~3–30%) of time is spent in the static "fully extruded" state where the loop is anchored by CTCF at both ends[14,15]. It remains unclear whether these two TADs are representative of loop extrusion dynamics across the human genome, what chromatin factors influence extrusion, and what role these actively extruding structures play in transcription.

To explore the genome-wide dynamic behavior of CTCF-anchored loops, we developed an analysis approach for 3D chromatin proximity ligation assays that enables simultaneous assessment of 3D contacts and fragment-level identification of CTCF/cohesin occupancy. We chose to use MNase digestion-based assays (MNase HiChIP[16–19], Region Capture Micro-C[12,20–22]) as the endo-exonuclease activity of MNase allows "footprinting" the location of bound factors[23–27]. We show that by analyzing proximity ligation events involving CTCF binding site locations, it is possible to use precise fragment size characteristics to determine at a single molecule level whether a given DNA molecule was bound by a nucleosome or CTCF/cohesin. Analysis of the CTCF/cohesin bound molecules then enables us to quantify the fully extruded (CTCF-CTCF loop) frequency at individual CTCF sites and find that it varies from ~1–10%. Further, we show that extruded loop length differs substantially depending on chromatin state, ranging from ~140 kb in active chromatin regions to ~250 kb in quiescent regions. These data advance our ability to quantify loop dynamics from topological data and support a role for partially extruded loops in gene regulation.

## Results

### MNase digestion fragment length distinguishes CTCF and nucleosome bound DNA

We used Micrococcal nuclease (MNase) HiChIP[16–19] for CTCF to profile 3D architecture in K562 cells, generating 150 bp reads with over 380 million unique pairwise contacts across four replicates. Briefly, following cell fixation with DSG and formaldehyde, chromatin is digested by MNase, immunoprecipitated to enrich for CTCF-bound DNA, and free ends are then ligated. After reverse-crosslinking, the resulting ligation products are sequenced from both ends and the mapping locations of the paired reads can be used to infer chromosomal locations of the physically interacting loci. We confirmed that the majority (70%) of loops identified by FitHiChIP[28] overlap those identified in a publicly available, high-coverage K562 Intact Hi-C dataset[29] (1.2 billion paired contacts) (see Methods, Supplementary Fig. 1).

We explored whether the MNase HiChIP assay could simultaneously be used to infer 3D contacts and the identity of proteins bound at interacting loci. MNase has previously been used to identify "footprints" of bound factors[23–26], as it has both endonuclease activity that selectively cleaves naked DNA not shielded by bound proteins, and exonuclease activity that subsequently trims back the remaining unprotected DNA fragment ends. We sought to estimate the protected DNA fragment length, as we hypothesized that this length would let us infer characteristics of a bound protein. This is possible in cases where an MNase digestion fragment is shorter than the read length such that the read comprises two separate segments and the ligation junction is directly observed (Fig. 1A, Supplementary Fig. 2). Exact fragment lengths could thus be determined for fragments with length less than 150 bp as the 150 bp read length results in censoring of longer fragments.

As expected, due to the high abundance of histones in chromatin (Fig. 1B), the predominant fragment length is approximately 150 bp or longer, suggestive of cuts between nucleosomes[30] (Fig. 1C). We also noted a distribution of shorter fragment lengths, with 20% of fragments representing lengths shorter than 120 bp (Fig. 1D). Restricting to fragments that overlap CTCF motifs shows a strong enrichment of

these short fragments (Fig. 1C), with <80 bp fragments having a 10-fold higher overlap frequency with CTCF motifs than >120 bp fragments (Fig. 1D).

A fragment pileup metaplot centered on CTCF motif loci (Fig. 1E) shows a strong enrichment of short fragments centered on the CTCF motif sequence, and a concomitant depletion of long fragments over the motif (Fig. 1F). Long fragments, in contrast, show peaks with a strong ~200 bp periodicity adjacent to the CTCF binding site (Fig. 1F), consistent with the ability of CTCF to precisely position a series of nucleosomes adjacent to its binding site[31]. Note that while long (>120 bp) fragments are depleted at CTCF binding sites, they still represent a significant fraction of reads at these sites (Fig. 1C, F). This likely reflects that CTCF motif loci without a bound CTCF are frequently instead occupied by histones[31,32], and even CTCF motifs with very strong CTCF ChIP-seq signal show only partial occupancy by CTCF across a cell population[33].

In summary, long (~150 bp) fragments likely correspond to nucleosome-protected DNA whereas shorter fragments at CTCF binding sites likely arise from CTCF-protected DNA. This can be explained by the different sizes of CTCF and histone octamers, which translate into different lengths of DNA protected from MNase exonuclease activity[31].

### Sub nucleosome-sized protected fragments enable precise CTCF binding site localization

In order to characterize CTCF-mediated chromatin interactions, we first set out to map CTCF binding sites with high resolution. We examined the MNase CTCF HiChIP fragment distribution around individual CTCF motifs and found a strong enrichment for short fragments (see methods) in the 150 bp windows centered on motifs (Fig. 2A, B). Further, a strand-specific analysis shows a bimodal read start position distribution centered on the CBS, with read 5' location peaks observed upstream (positive strand) and downstream (negative strand) of the CBS (Fig. 2C, D). We refer to these regions as quadrants 2 and 4 (Q2 and Q4) respectively (Fig. 2D, E). This suggests that CTCF binding can be detected by an enrichment of reads in Q2 and Q4 relative to reads in Q1 and Q3 (Fig. 2D, E). This pattern was observed consistently across four replicates (Supplementary Fig. 3A). At sites without protein binding, MNase can cut at any location resulting in no specific quadrant enrichment (Fig. 2E). As a result, we attempted to determine binding events by testing if there are significantly more reads in Q2 and Q4 than Q1 and Q3 (Fig. 2F).

Statistically we assess each genomic location by considering each adjacent read in the 75 bp upstream and 75 bp downstream region as an independent draw from a multinomial distribution with four categories corresponding to the four quadrants. Under the null hypothesis, each read has equal probability of belonging to any of the four quadrants $Q_i, i \in \{1, 2, 3, 4\}$. Because CTCF binding induces a strong preferential read pile-up in _both_ quadrants 2 and 4 (Fig. 2C–E), we test for an enrichment of reads in Q2 and Q4 compared to Q1 and Q3 by estimating the *CAMEL* statistic $\hat{\alpha} = \frac{\min(n_2, n_4)}{\max(n_1, n_3)}$, where $n_i$ is the number of reads in $Q_i$. We then test if $\hat{\alpha}$ is significantly greater than 1. Note that min and max are used to enforce that both quadrants 2 and 4 must have more reads than both quadrants 1 and 3; using the average enables spurious read pile-ups that occur in quadrant 2 or 4 (but not both) to be called as false positive CTCF binding events. Nominal p-values are computed using an empirical null distribution generated through multinomial sampling (see methods).

We sought to evaluate this CTCF binding site identification approach using CTCF motif locations[34], CTCF ChIP-seq peaks[29], and DNA loop anchors identified by FitHiChIP at 2.5 kb resolution[28]. We first defined a high stringency true positive set of CTCF binding sites as CTCF motifs in loop anchors that are located within 30 bp of a CTCF ChIP-seq peak summit. To avoid ambiguity due to multiple closely spaced motifs, we further selected only those motifs that are unique

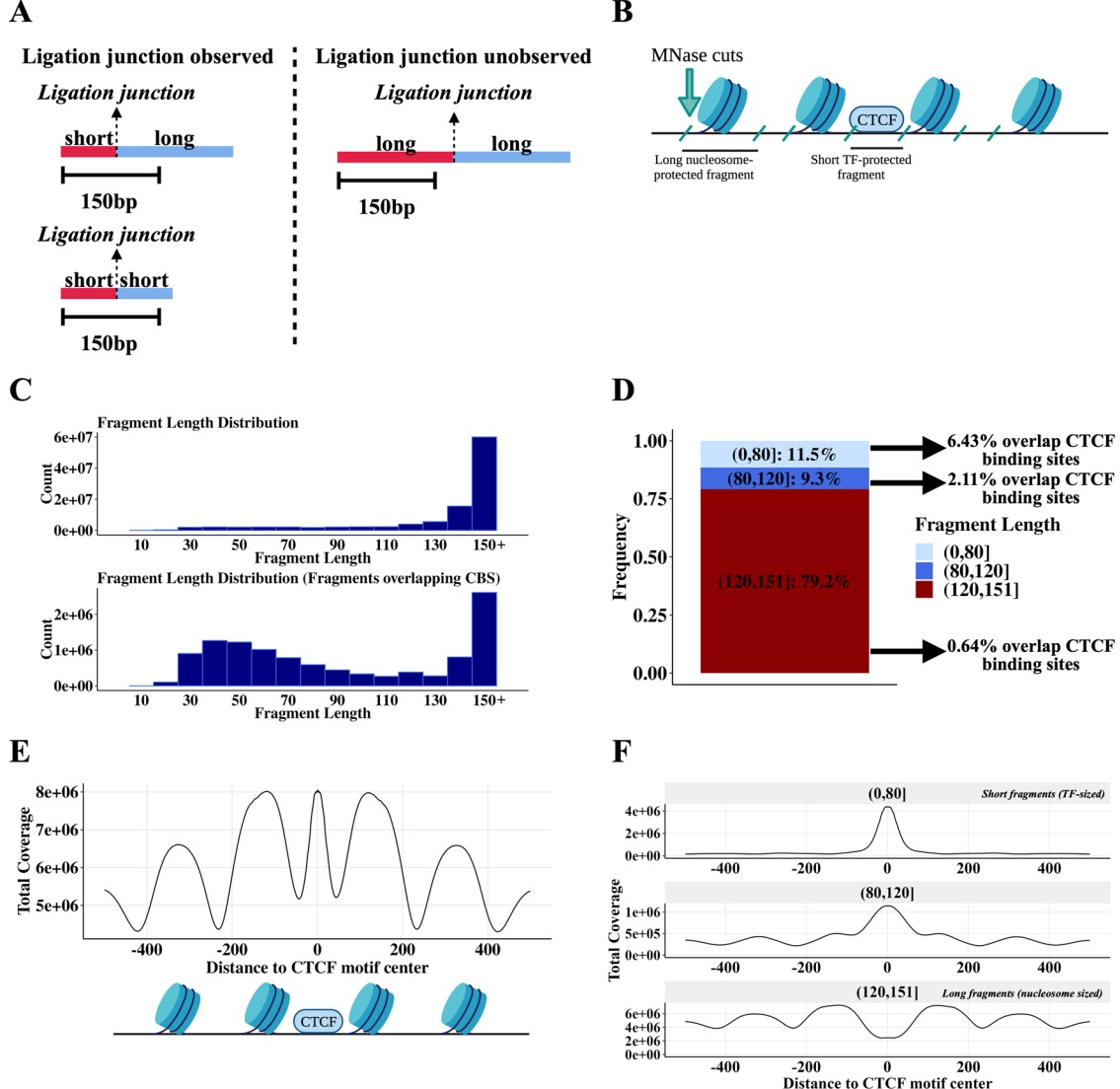

**Fig. 1 | MNase CTCF HiChIP data contains short (~<80 bp) CTCF-protected fragments and longer (~>120 bp) nucleosome-protected fragments.**
**A** Schematic illustrating relationship between short fragments and observed ligations. **B** Schematic illustrating how the fragment length results from MNase cutting around bound proteins of different sizes. Created in BioRender. Aryee, M. (2025) https://BioRender.com/g98u240 **C** Fragment length distribution for all fragments (top plot) and fragments overlapping occupied CTCF motifs (lower plot). Occupied CTCF motifs are defined here as CTCF motifs within 30 bp of a CTCF ChIP-seq peak summit. **D** Bar plot quantifying the frequency of different fragment lengths genome-wide and how often each fragment length group overlaps an occupied CTCF motif. Occupied CTCF motifs are defined here as CTCF motifs within 30 bp of a CTCF ChIP-seq peak summit. **E** Fragment coverage metaplot +/− 500 bp around CTCF binding sites. Schematic below the coverage metaplot illustrates the likely proteins producing these peaks. Schematic created in BioRender. Aryee, M. (2025) https://BioRender.com/g98u240 (**F**) Plot (**E**) stratified by fragment length.

within a 2.5 kb loop anchor. Using this true positive set, we observe that the *CAMEL* statistic, $\log 2(\hat{\alpha}) = \log 2(\frac{\min(n_2, n_4)}{\max(n_1, n_3)})$ has signal greater than 0 (equivalently, $\hat{\alpha} > 1$) almost exclusively within 20 bp of the CTCF motif center and centered on 0 bp from the CTCF motif center (Fig. 3A, Supplementary Fig. 3B). Using this same set of true positive sites (true negatives are the regions of the loop anchors >200 bp from a CTCF motif), we achieve >90% precision and >90% recall at a nominal *p*-value threshold of 1e-05, and maintain high recall and precision at all *p*-value thresholds <1e-05 (Fig. 3B, Supplementary Fig. 3C). This high level of recall and precision is achieved because of the very different *CAMEL* statistic distributions for true positives and true negatives (Fig. 3C).

Running *CAMEL* binding site detection genome-wide, we observe that almost all peaks (93%) are within 20 bp of a CTCF motif center, with a median distance of 5 bp (Fig. 3D). Defining accuracy as motif occurrence within 20 bp of a peak summit, we find that *CAMEL* maintains ~95% motif occurrence (Fig. 3E). Further, applying the motif

discovery tool STREME[35] to 30 bp sequences centered on *CAMEL* peak summits produces a motif sequence that exactly matches the JASPAR CTCF motif (Fig. 3F), supporting *CAMEL*'s ability to identify true CTCF binding sites.

## Short, TF-scale fragments are enriched for long-range interactions

We next examined the length characteristics of MNase HiChIP fragments overlapping individual CTCF motifs, in an attempt to infer the identity of the proteins occupying each locus. For all motifs with non-zero coverage, we observed long, 150+ bp fragments, as shown for two representative motifs in Fig. 4A. We hypothesized that these fragments represent cells with a nucleosome located at the motif locus. In addition, particularly for CTCF motifs with an overlapping CTCF ChIP-seq peak, we also observed short, sub-nucleosome sized (<120 bp) fragments (Fig. 4A, left). The fraction of reads overlapping a CAMEL-

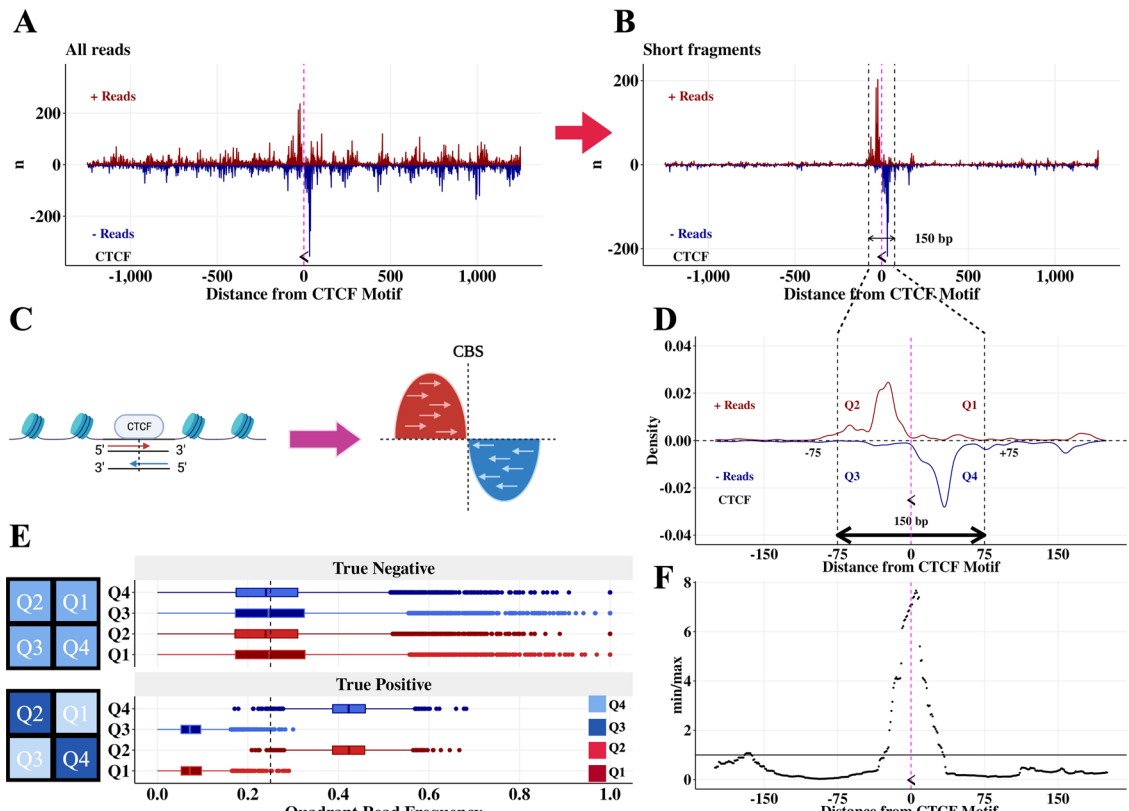

**Fig. 2 | True CTCF binding sites have a bimodal strand-specific distribution centered on the CTCF motif. A** Unfiltered reads +/− 1250 bp around a CTCF binding site located on the negative strand (chr1: 30,779,763 − 30,779,781). The midpoint of the CTCF motif is marked with the symbol " < ", representing that it is on the negative strand, and a pink line. **B** Plot (**A**) filtered to observed ligations (equivalently, short fragments.) **C** Schematic demonstrating the bimodal read pile-up around a CTCF binding site. Created in BioRender. Aryee, M. (2025) https://BioRender.com/g98u240 (**D**) Plot (**B**) as a density plot and zoomed in on the CTCF motif, with quadrant annotations. **E** Distributions of reads in quadrants for true negative and true positive CTCF binding sites in DNA loop anchors. True positives are defined as CTCF motifs that are the only CTCF motif in a loop anchor and within 30 bp of a CTCF ChIP-seq peak. True negatives are areas of the loop anchors with one CTCF motif that are at least 200 bp from the CTCF motif. 3,984,150 fragments across 4523 loop anchors each containing one CTCF binding site are used to make these boxplots. Boxplots are made with ggplot2::geom_boxplot() and show the 25% quantile (lower bound of box), median (center line), and 75% quantile (upper bound of box). Boxplot whiskers subtract (lower whisker) or add (upper whisker) 1.5 * IQR from the lower bound (lower whisker) or upper bound (upper whisker) of the box. Schematics of the quadrant read pile-up patterns are shown next to the corresponding true positive and true negative boxplots. **F** *CAMEL* statistic ($\hat{\alpha} = \frac{\min(n_2, n_4)}{\max(n_1, n_3)}$) for plot (**D**) peaks at the CTCF motif.

identified CBS (+) that are short (<120 bp) strongly correlates with the strength of its CTCF ChIP-seq signal (Fig. 4B).

Given the association between short fragment frequency and binding of the architectural protein CTCF, we asked whether short and long fragments might have different long-range interaction behavior. We stratified the 2.3 million HiChIP contacts anchored at CAMEL-identified CBS (+) by the size of the motif-overlapping fragment and examined the distance to the fragments' interaction partner. This reveals that interactions that span linear distances greater than 10 kb are significantly enriched (2.2-fold, $p < 10^{-10}$ (chi-square)) for short TF-scale fragments (Supplementary Fig. 4A). Consistent with this, HiChIP contact maps created using only <120 bp fragments more focally highlight CTCF-mediated contacts (Supplementary Fig. 5).

**CTCF and Cohesin occupancy footprints**

Focusing on the short, CBS-overlapping fragments (<120 bp), we found that they exhibit a bimodal interaction length distribution (Supplementary Fig. 4A), with 39% involving an interaction partner >10 kb away. We sought to understand factors that might explain this long- vs short-range interaction behavior through a closer examination of the fragment characteristics. To this end we created aggregate 2D density "footprint plots" showing fragment positions relative to CTCF motifs at CAMEL-identified CTCF binding sites (+) by plotting fragment start

positions (x-axis) against fragment end positions (y-axis) (Fig. 4C), such that each (fragment start, fragment end) combination represents a single point. Fragment start and fragment end have been aligned to the start of the 35 bp JASPAR CTCF motif (MA1930.1) such that 0 represents the start of the motif. Examining first only those fragments with short-range (<10 kb) interactions (Fig. 4C left) we noted that two fragment footprint types were most prominent. The footprints correspond to two different fragment types, with a shared end position but differing starts. To interpret these footprints we overlaid them onto the CTCF motif (MA1930.1). The motif contains a core 19 bp region that is present at the majority of CTCF binding sites genome-wide, and an upstream 10 bp region that is present at a smaller subset of CTCF sites[36,37] (Fig. 4D, Supplementary Fig. 4B). Strikingly, we find that the boundaries of the footprints correspond almost precisely to two different binding events: one that protects the 19 bp core region only and another binding event that protects the full 35 bp core + upstream region (Fig. 4C, D).

We next examined footprint plots for the short TF-scale fragments (<120) with long-range (>10 kb) interaction partners (Fig. 4C right). These fragments also comprise two distinct types with the same start positions as the short-range interaction fragments, corresponding to the beginning of either the core or upstream CTCF motif region. Interestingly however, both the

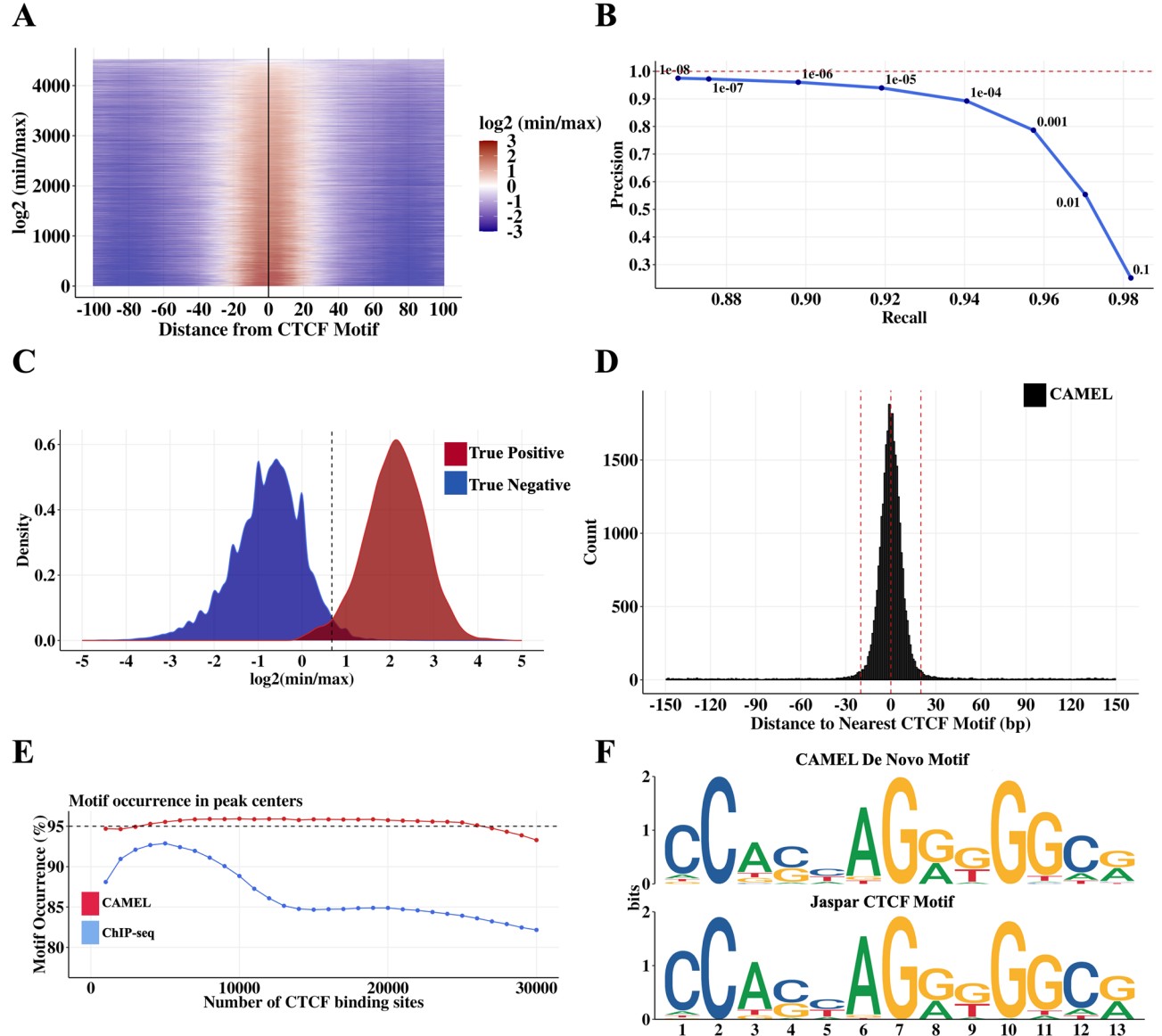

**Fig. 3 | CTCF binding sites identified by *CAMEL* with single basepair resolution in MNase K562 CTCF HiChIP data. A** Heatmap of log2(min/max) as a function of distance between *CAMEL* peak center and CTCF motif center within loop anchors. Only CTCF motifs that are unique within a loop anchor and within 30 bp of a CTCF ChIP-seq peak are used. **B** Precision recall curve for true negative and true positive CTCF binding sites in DNA loop anchors. True positives are defined as in (**A**). True negatives are areas of the loop anchors in (**A**) that are at least 200 bp from the one CTCF motif. Precision is calculated as TP / (TP + FP), recall is calculated as TP / (TP + FN). **C** *CAMEL* statistic $\log 2(\hat{\alpha}) = \log 2(\frac{\min(n_2, n_4)}{\max(n_1, n_3)})$ density plots using the same set of true positives and true negatives as (**B**). **D** Histogram with 1 bp bin size depicting *CAMEL* resolution for all peaks genome-wide (not just in loop anchors). **E** Motif occurrence in ChIP-seq and *CAMEL* peak centers genome-wide. Motif occurrence is calculated as % peak centers within 20 bp of CTCF motif. Only peak centers within 150 bp of a CTCF motif are used for this figure. **F** 30 bp sequences centered on genome-wide *CAMEL* peak centers produce a de novo motif (top) that matches the core JASPAR CTCF motif (bottom). De novo motif is made using STREME[35].

Core and Core+Upstream fragment types have an extended downstream end that stretches ~15 bp beyond the CTCF footprint (Fig. 4C, D, Supplementary Fig. 4C), suggesting the possible presence of an additional protein protecting this region. CTCF binds its motif in an orientation that presents the N-terminus towards the downstream end, and recent work has shown that regions within this N-terminus interact directly with the loop-extruding cohesin complex[6,38]. Based on this and the link between the CTCF-downstream footprint and long-range interactions, we hypothesized that the footprint indicates cohesin's presence.

To investigate whether the CTCF downstream extended footprint reflects cohesin occupancy we turned to a degron strategy that achieves 97% cohesin depletion 3 hours after auxin (IAA) induction[20].

Goel et al. have previously used this system with Region Capture Micro-C[20], a procedure involving a similar MNase digestion to MNase HiChIP, to profile 3MB of DNA containing 65 CTCF binding sites at the *Fbn2*, *Klf1*, and *Ppm1g* loci in mouse embryonic stem cells. We resequenced these RCMC libraries from the IAA-induced cohesin degron and DMSO control conditions[20] using 150 bp paired end reads to allow accurate assessment of fragment length. Examining fragment footprint plots for short TF-scale fragments we observe a 15 bp extended footprint downstream of the CTCF motif in the DMSO control condition that is greatly attenuated upon cohesin depletion by IAA (Fig. 4E, F —red box).

Taken together, these findings suggest that base-pair resolution analysis of MNase digestion fragment start and end positions at CTCF

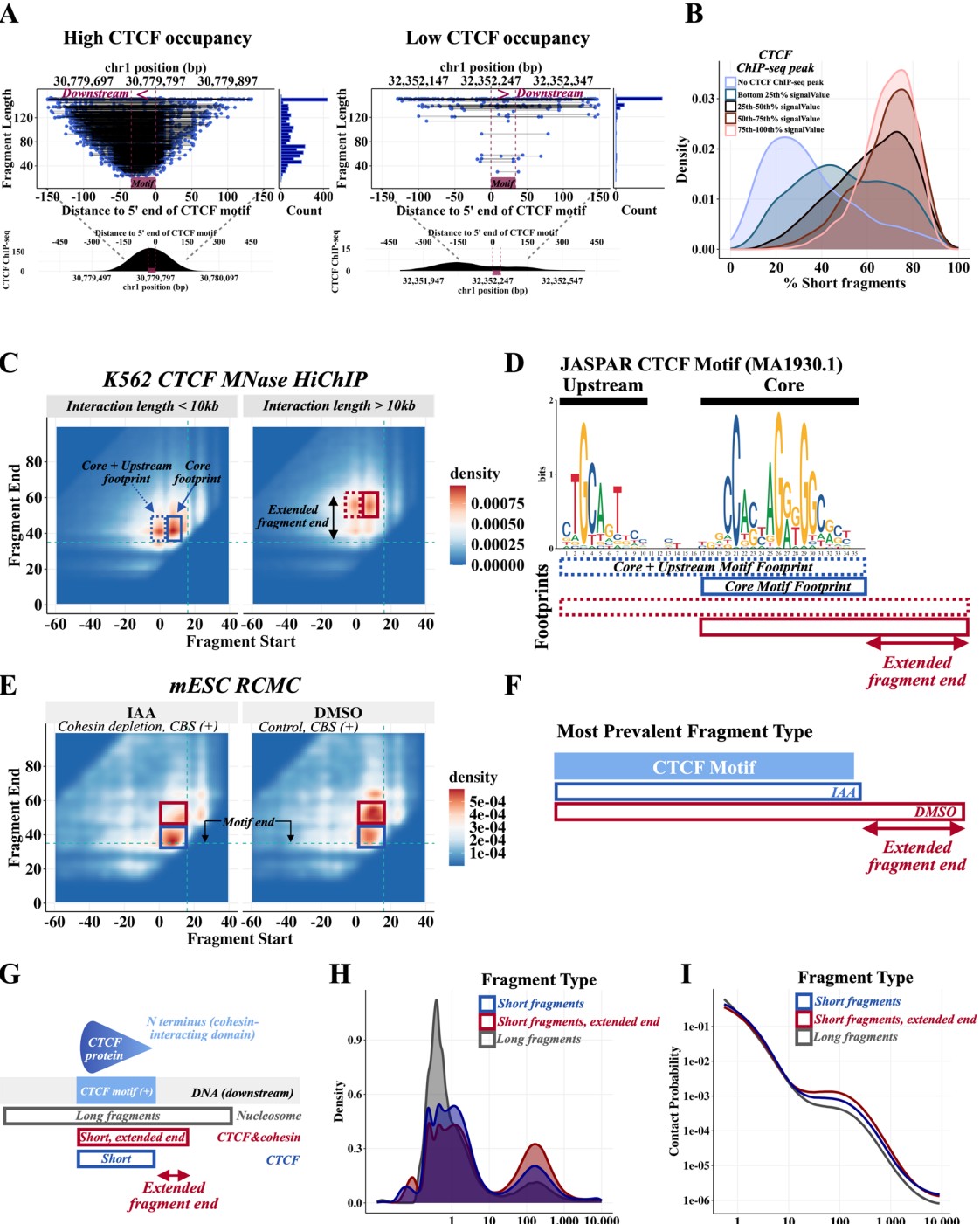

**Fig. 4 | Cohesin footprint observed in MNase HiChIP and experimentally confirmed in mESC RCMC. A** High and low CTCF occupancy motifs. For each occupancy level, CTCF ChIP-seq and all fragments overlapping the CTCF motif are depicted, along with the corresponding fragment length histogram. **B** The fraction of reads overlapping a CBS (+) that are short (<120 bp) strongly correlates with the strength of its CTCF ChIP-seq signal. The fraction of short reads (<120 bp) was calculated for each CBS, plotted in a density, and colored by absence of CTCF ChIP-seq peak or CTCF ChIP-seq signalValue quartile. **C** K562 CTCF MNase HiChIP 2D density for short (<120 bp) left fragments overlapping CAMEL-identified CBS (+) (N = 10,906 CBS). The fragment start and fragment end are aligned relative to the start of the 35 bp CTCF motif (MA1930.1), so that "0" in the plots corresponds to the start of the 35 bp CTCF motif. **D** Annotated 35 bp JASPAR CTCF motif (MA1930.1).

**E** mESC RCMC 2D density for short (<120 bp) left fragments overlapping CTCF motifs within 30 bp of a mESC CTCF ChIP-seq peak (N = 65 CBS). The fragment start and fragment end are aligned relative to the start of the 35 bp CTCF motif (MA1930.1), so that "0" in the plots corresponds to the start of the 35 bp CTCF motif. **F** Schematic illustrating the most prevalent fragment type observed in the mESC RCMC perturbations shown in (**E**). **G** Schematic showing the three classified types of fragments relative to the location of the CTCF motif. The interaction partners of these three fragment types are graphed in (**H**, **I**). **H** Short TF-sized fragments (<120 bp) with an extended fragment end have a noticeably larger bump in density of long range interactions (>10 kb) compared to TF-sized short fragments without an extended fragment end and nucleosome-sized long fragments (>120 bp). **I** P(S) curve for fragments depicted in (**H**).

binding sites can classify individual fragments as likely bound by a nucleosome, CTCF alone, or CTCF together with cohesin. Given cohesin's role in mediating DNA loops, we hypothesized that cohesin-associated DNA would be more likely to participate in long-range chromatin interactions. Using the proxy of a CTCF-downstream cohesin footprint to identify likely cohesin-protected DNA fragments, we indeed observe that these fragments have increased propensity for long-range interactions in the 10 kb–500 kb range compared to DNA bound to CTCF alone (Fig. 4G–I).

**Active chromatin obstructs cohesin-mediated loop extrusion**
Using the techniques described above, MNase proximity ligation assays enable us to simultaneously locate CBS at high resolution, identify footprints of bound proteins, and interrogate specific chromatin contacts at the single molecule level. We next used these data to identify cohesin-protected DNA fragments and characterize cohesin extrusion dynamics in a range of genomic contexts.

We first sought to determine the genome-wide frequency of fully extruded (CTCF-CTCF) loops, which have been previously estimated using locus-specific live-cell imaging studies[14,15]. We used K562 MNase HiChIP data to estimate the fully extruded rate for each CAMEL-identified CTCF binding event by selecting short fragments overlapping the CBS with an extended fragment end (Fig. 4C, Supplementary Fig. 4C) and determining the fraction of interaction partners that overlap a downstream convergent CTCF motif (see methods). We find wide CBS to CBS variability with fully extruded rate estimates ranging from ~1 to 10% (Fig. 5A) and a genome-wide average frequency of 5%. These findings are compatible with an imaging study of the 505 kb *Fbn2* TAD that estimated a fully extruded loop frequency of 3%–6%[14], and suggest that the majority of CTCF-anchored chromatin contacts genome-wide are in the 'extruding' state, rather than joining two CTCF sites. We note that these fully extruded loop estimates are contingent on at least one end of a long-range interaction being anchored by CTCF, and does not include long-range interactions that are not anchored by CTCF on either end (the fully unlooped state).

We next sought to use our data to examine how cohesin extrusion is impacted by chromatin context. Since HiChIP libraries are a snapshot of millions of cells, we can estimate dynamic extrusion parameters such as the average loop size extruded by cohesin[39] from interaction length distributions. To determine the impact of chromatin state on cohesin extrusion, we first annotated the 1 MB region downstream of each *CAMEL*-identified CBS with ChromHMM chromatin states[40] (Fig. 5B, see methods) to characterize the DNA through which a cohesin anchored at the CBS would extrude. We collapsed ChromHMM states into three main categories: active, polycomb/bivalent or quiescent (Supplementary Fig. 6A), and computed the fraction of downstream chromatin in each category. Each of the three chromatin categories was represented by the 20% of regions with the highest fraction of DNA in this state. We used CTCF/cohesin-protected fragments (Fig. 4C, Supplementary Fig. 4C) overlapping the three sets of CBS and estimated extruded loop sizes using fragment-level interaction lengths.

Interestingly, we find that cohesin loop sizes are ~1.75 fold greater in quiescent regions (~250 kb) than active regions (~140 kb), corresponding to a difference in average extruded loop size of ~110 kb, $p < 10^{-10}$ (Fig. 5C, Supplementary Fig. 6B). The P(s) curve, a plot of interaction frequency vs. distance, confirms a depletion of the longest-range interactions in active regions (Fig. 5D). This ~250 kb estimate for loop size in quiescent regions is consistent with a live cell imaging study of the *Fbn2* locus in the absence of transcription that estimated an average loop size of 300 kb[14]. As quiescent regions are characterized by low TF binding and low transcription[41], we hypothesized that the substantial difference in extruded loop size might relate to gene

activity and enhancer density obstructing loop extrusion. Consistent with this, we found that higher levels of H3K27ac and RNA Pol II binding in the 1 MB region downstream of the CBS strongly correlate with lower average extruded loop size (Fig. 5E). In addition, a focal enrichment of contacts between CBS and downstream H3K27ac-marked enhancers (Supplementary Fig. 6C) suggests an ability of these regulatory elements to stall extrusion.

We used the mESC RCMC cohesin degron system to assess the cohesin dependency of contact enrichment between CTCF binding sites and regulatory elements. The captured region contains the *Fbn2* gene and its encompassing convergently oriented CTCF binding sites (Supplementary Fig. 7A). A pseudo 4 C plot showing interactions anchored at the *Fbn2* upstream CTCF viewpoint confirms the presence of the fully extruded 505 kb TAD-defining CTCF-CTCF loop and also reveals a shorter range contact enrichment between the CTCF anchor and the 230 kb downstream RNAPII-marked *Fbn2* transcriptional start site (TSS) (Supplementary Fig. 7B). Consistent with this, the TSS is also marked by cohesin (RAD21 & SMC1A) ChIP-seq peaks. As expected, cohesin depletion significantly weakens the fully extruded CTCF-CTCF TAD contacts. Importantly, the loss of cohesin also attenuates the 3D contacts between the CTCF anchor and the TSS, suggesting that these contacts are also cohesin-dependent. These findings further suggest that regulatory elements are able to impede cohesin extrusion, and are consistent with recent reports of RNAPII/cohesin interactions[42,43].

We performed two additional sets of analyses to confirm that the estimated differences in loop extrusion length between chromatin states are not confounded by background, non cohesin-related, interactions. First, since each genomic region has locus-specific genetic and epigenetic architecture, we fit a linear mixed effects model to account for the locus-to-locus variability in interaction patterns (see methods). Specifically, we compute the 'cohesin effect' on loop length, defined as the average increase in interaction length for CTCF/cohesin bound fragments while adjusting for the interaction length of nucleosome bound fragments at each individual CBS. Controlling for the background interaction frequency due to the local sequence and epigenetic state context of a region in this way confirms that cohesin-associated loops are significantly shorter in active chromatin (see Methods and Supplementary Fig. 6D). Second, we examined whether differences in CTCF binding density between active and quiescent chromatin might account for the observed differences in average extruded loop size. We repeated the loop length analyses after excluding contacts between convergent CTCF motifs and confirmed that the relationship between chromatin state on loop length remains largely unaltered (Supplementary Fig. 6E, F).

**Discussion**
While chromatin loops as visualized on proximity ligation contact maps are often viewed as static, it is evident from recent single locus live cell imaging studies that these structures can be very dynamic. Here we provide a genome-wide view of extrusion dynamics by using MNase-based proximity ligation data to simultaneously infer 3D genome structure and the presence of the architectural factors CTCF and cohesin on individual DNA molecules. The dual endo- and exo-nuclease activity of MNase degrades unprotected DNA up to the edge of bound proteins resulting in fragment sizes that are characteristic of either nucleosome or CTCF/cohesin binding. Our findings demonstrate that an analysis of precise fragment start and end positions enables the classification of individual DNA molecules participating in 3D interactions based on their association with either nucleosomes or CTCF. Further, a subset of CTCF-bound molecules have an additional extended footprint immediately

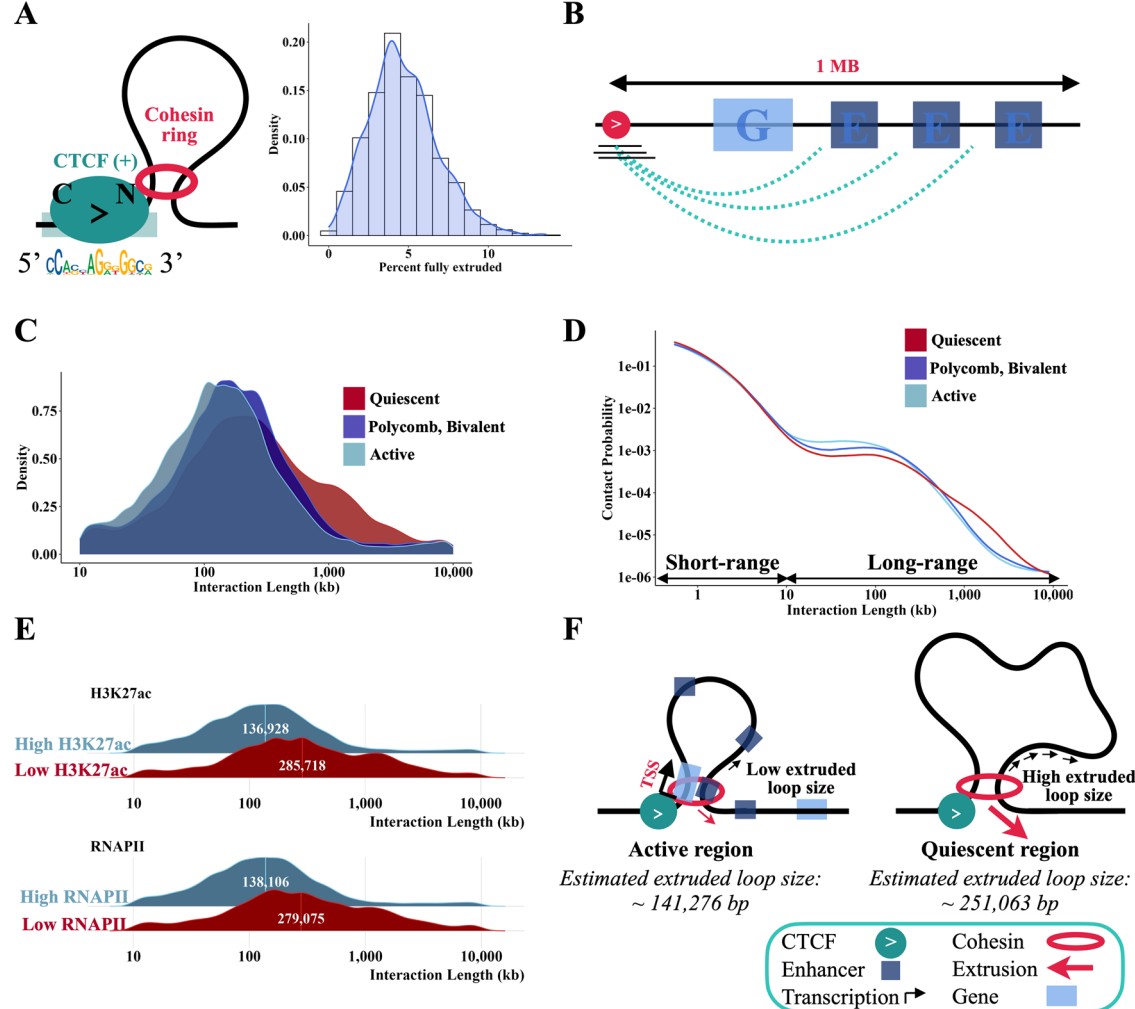

**Fig. 5 | Cohesin extrudes further through quiescent regions than active regions. A** Most CTCF-mediated looping contacts do not reflect the fully extruded state. Estimate is obtained using left fragments that overlap *CAMEL* identified CBS (+) and have an interaction length greater than 10 kb with start and end at least 5 bp from motif start and end, length <120, and extended fragment end. For each CBS with at least 50 long-range TF-protected fragments overlapping the motif, % convergent is calculated as the number of interaction partners overlapping CTCF (-) motifs / total number of fragments at motif. Because this estimate is conditional on CTCF binding at the anchor, we divide estimates by two to account for the ~50% occupancy of CTCF[33]. **B** Depiction of how 1 MB regions downstream of CBS were annotated using ChromHMM. Density (**C**) and P(S) curves (**D**) for chromatin state clusters shown in (Supplementary Fig. 6A), filtered to the top 20%. Chromatin annotations making up each cluster are added together and quantiles are obtained

to determine fragments in the top 20% of active chromatin, quiescent chromatin, and bivalent / polycomb chromatin. Left fragments that overlap *CAMEL* identified CBS (+) and have an interaction length greater than 10 kb with start and end at least 5 bp from motif start and end, length <120, and extended fragment end are used. **E** Ridge plots for the bottom 10% quantile ("Low") and top 10% quantile ("High") of H3K27ac bp and number of RNAPII binding sites. ChIP-seq from ENCODE was used to annotate 1 MB downstream of left fragments overlapping CBS (+) for this figure. Left fragments that overlap *CAMEL* identified CBS (+) and have an interaction length greater than 10 kb with start and end at least 5 bp from motif start and end, length <120, and extended fragment end are used. Plots are labeled with the average log10 interaction length. **F** Diagram illustrating differences in extrusion rates between active and quiescent chromatin states, with numbers obtained from Supplementary Fig. 6B.

downstream of the CTCF binding site that can be attributed to the co-presence of a cohesin complex. This is consistent with a co-crystal structure of CTCF and the SA2-SCC2 cohesin subunits that shows cohesin docking at the N-terminus of the CTCF protein, positioned at the downstream edge of the motif.

By leveraging this 'footprinting' property of MNase together with a strand-specific analysis of digestion fragment sizes, we show that MNase CTCF HiChIP can be used to identify CTCF binding sites with 5 bp precision in K562 cells, on-par with the precision of the dedicated localization assay CUT&RUN[25]. This precision allows the discrimination of bound and unbound CTCF motifs located in close proximity. Since loop anchors often contain multiple CTCF motifs (Supplementary Fig. 6G), not all of which are bound, distinguishing between occupied and unoccupied CTCF motifs at high resolution is vital for determining the CBS mediating a chromatin loop. We have implemented this

approach in an accessible software package, *CAMEL* (CTCF Analyzer with Multinomial Estimation, https://github.com/aryeelab/cohesin_extrusion_reproducibility).

We use the ability to identify individual 3D contacts anchored by CTCF and/or cohesin to make several inferences about cohesin extrusion dynamics genome-wide. We estimate the frequency with which a CTCF bound locus forms a loop with a downstream CTCF site and find that it varies considerably from CBS to CBS, with a genome-wide range from ~1 to 10%. This is consistent with estimates from two recent live-cell imaging studies that found that CTCF-mediated loops predominantly exist in the partially extruded state at two imaged TADs[14,15]. We note a limitation of our study in that the use of HiChIP may lead to an over-estimate of the fully extruded rate.

We next explored the impact of chromatin state on loop extrusion and observed an approximately 2-fold increase in

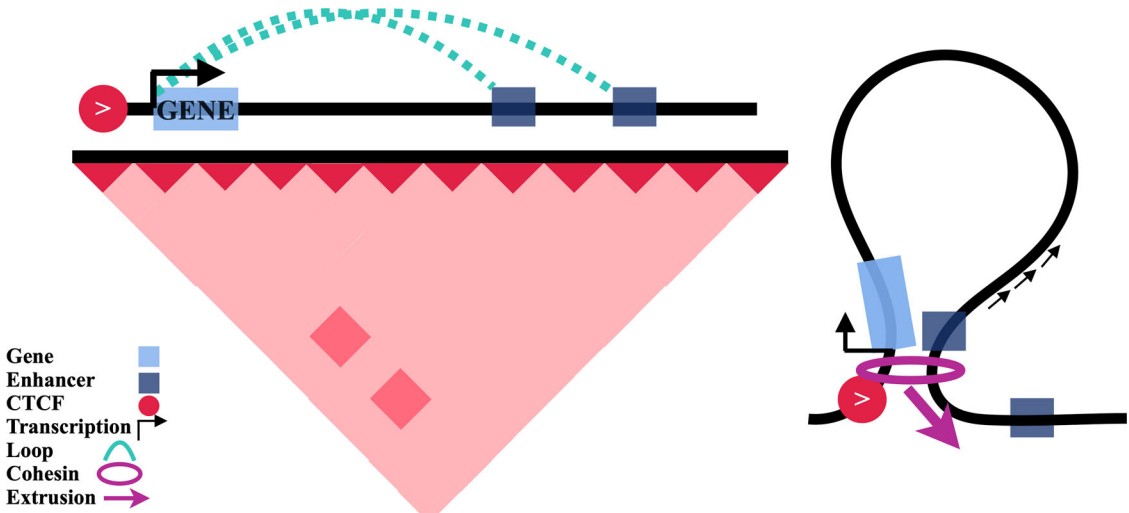

**Fig. 6 | Schematic of supported model whereby single promoter-proximal CTCF sites enable an enrichment of enhancer-promoter contacts.** Left: contact map representation. Right: physical loop representation of left panel.

extruded loop size comparing quiescent chromatin to active chromatin, an effect that correlates with differences in H3K27ac and RNAPII levels between these chromatin states. Consistent with this, we also observe punctate contact enrichment between CTCF bound sites and downstream H3K27ac-marked enhancers and RNAPII-marked transcriptional start sites. Analysis of a cohesin degron system revealed that these contacts are attenuated upon cohesin depletion. This is in line with previous ChIP-seq studies showing co-localization of RNAPII and cohesin in the absence of CTCF[44], despite evidence against preferential loading of cohesin at regulatory elements[42]. These findings are consistent with a model of stalled extrusion at regulatory elements and previous studies suggesting that gene and enhancer activity obstruct cohesin activity[45–50]. A limitation of our study, however, is that our data do not allow us to exclude alternate reasons such as differences in extrusion speed or cohesin unloading activity between active and less active chromatin regions.

This dynamic CTCF-mediated enhancer-promoter contact model is consistent with recent evidence that cohesin-mediated chromatin loops are dynamic[14,15] and that promoter proximal CTCF binding sites facilitate contacts with distal enhancers[10–12,50,51]. Our findings are also compatible with the "kiss and kick" model of gene regulation where enhancer-promoter contacts are temporarily maintained while RNAPII is paused at transcriptional start sites, and are subsequently lost during elongation[52].

In summary, these findings suggest that the concept of chromatin loops should be expanded beyond a static structure to include an actively extruding state, where loops anchored by a single CTCF may play a key role in enabling the 'scanning' of DNA for regulatory elements (Fig. 6).

## Methods
### Data generation
**K562 (ATCC CCL-243) CTCF MNase HiChIP.** Four MNase K562 CTCF HiChIP (150 bp paired-end) libraries were generated using the Cantata Bio / Dovetail Genomics MNase HiChIP kit[17–19]. CTCF MNase HiChIP was performed as described in the Dovetail HiChIP MNase Kit protocol v.2.0. Briefly, 5 million K562 cells per sample were crosslinked with 3 mM DSG and 1% formaldehyde and digested with 1ul MNase ("YET" samples) or 2 ul MNase ("GW" samples) in 100 ul of 1X nuclease digestion buffer. Cells were lysed with 1X RIPA containing 0.1% SDS, and CTCF ChIP was performed using 1500 ng of chromatin (40-70%

mononucleosomes) and 500 ng of CTCF antibody (Cell Signaling, cat #: 3418). Protein A/G beads pull-down, proximity ligation, and library preparation were done according to the protocol. Libraries were sequenced to a read depth of -172 million paired end reads per sample on the Illumina Nextseq 2000 platform.

**mESC (RRID: CVCL_J962) RCMC.** WT, 3 hour DMSO, and 3 hour IAA libraries (two replicates of each) at the *Fbn2* (chr18:57,899,000-58,901,000), *Ppm1g* (chr5:31,100,000-32,225,000), and *Klf1* (chr8:84,120,000-85,130,000) loci from Goel et al.[20] were re-sequenced with 150 bp paired-end reads. Note that the provided coordinates of the *Fbn2*, *Ppm1g*, and *Klf1* captured loci use the mm10 reference genome.

### Data Pre-processing
Four replicates of K562 MNase CTCF HiChIP and two replicates each of mESC RCMC WT, mESC RCMC DMSO, and mESC RCMC IAA were aligned to the reference genome (hg38 and mm10 respectively) using the BWA-MEM algorithm[53] (with *-5SP -T0*). Ligation events were then recorded using pairtools parse v. 0.3.0[54] (with *--min-mapq 40 --walks-policy 5unique --max-inter-align-gap 30 --add-columns pos5,-pos3,dist_to_5,dist_to_3,read_len*). Parsed reads were then sorted (pairtools sort) and PCR duplicates were removed (pairtools dedup). Final pairs and bam files were then generated using *pairtools split --nproc-in 8 --nproc-out 8 --output-pairs ${filename}.mapped.pairs --output-sam -| samtools view -bS -@16 | samtools sort -@16 -o ${filename}.-mapped.PT.bam;samtools index ${filename}.mapped.PT.bam* We used a Terra pipeline to keep data processing consistent.

Replicates were merged using pairtools merge and for the RCMC samples we then filtered to both fragments of an interaction pair in the captured regions using pairtools select (captured regions (mm10): *Fbn2 (chr18:57,899,000-58,901,000), Ppm1g (chr5:31,100,000-32,225,000), and Klf1 (chr8:84,120,000-85,130,000)).* For example, pairtools select '(chrom1 == "chr18") and (chrom2 == "chr18") and (57899000 <= pos1 <= 58901000) and (57899000 <= pos2 <= 58901000)' RCMC_WT.mapped.pairs > RCMC_WT_Fbn2.mapped.pairs.

CTCF MNase HiChIP loop calls were then made from the combined replicate pairs file using FitHiChIP Peak to Peak[28] with 2.5 kb loop anchor bin size. Balanced contact maps were created using the merged replicate genome-wide pairs file (K562 CTCF MNase HiChIP) and the merged replicate captured

region pairs files (mESC RCMC). Filtering to both fragments in the captured window for RCMC data processing is the approach taken by the RCMC authors to avoid problems with ICE balancing[20]. To create .mcool files, we compressed the combined replicate pairs files (bgzip), indexed the compressed pairs files (pairix), created cool files (cooler cload pairix), and then obtained ICE-balanced mcool files (cooler zoomify --balance). This data processing approach was developed starting with guidelines from https://hichip.readthedocs.io/en/latest/before_you_begin.html.

The fragment start and fragment end of each interaction partner are obtained by taking the minimum of the 5' and 3' ends of a fragment (this is the fragment start) and the maximum of the 5' and 3' ends of a fragment (this is the fragment end). Including --add-columns pos5,pos3 in the pairtools parse command adds the 5' and 3' ends of both fragments to the pairs file and is crucial for identifying fragment starts, fragment ends, and fragment lengths for both fragments in an interaction pair. Reproducible code is available at https://github.com/aryeelab/cohesin_extrusion_reproducibility.

Fragment lengths can be determined for fragments with length less than 150 bp; the 150 bp read length results in censoring of fragments longer than 150 bp. Since at least ~25 bp are required to align a sequence to the reference genome, we are able to identify both mapping locations for a ligation event when the longer fragment is <=125 bp. We use this as a proxy for identifying short fragments in Figs. 2 and 3. The fraction of informative, sub-nucleosome fragments decreases with shorter sequencing read length (Supplementary Fig. 2).

### Visualizing contact maps in R

Contact maps were visualized on the hg38 (human) and mm10 (mouse) reference genomes using HiContacts::plotMatrix after importing a given resolution of the mcool file using HiCExperiment::import. Publicly available bigwig files (ChIP-seq and ATAC-seq) were imported using rtracklayer::import, visualized with ggplot, and aligned with contact maps using cowplot::plot_grid. Downloaded mESC bigwig files were CrossMapped[55] from mm9 and mm39 to the mm10 reference genome. Downloaded K562 bigwig files were already aligned to hg38, and thus did not require CrossMapping.

Downloaded mESC files include: mESC CTCF ChIP-seq (GSM3508478), mESC ATAC-seq (GSE98390), mESC H3K27ac ChIP-seq (GSM2417096), mESC RNAPII ChIP-seq (GSM6809981), mESC RAD21 ChIP-seq (GSE137272), and mESC SMC1A ChIP-seq (GSE137272). Downloaded K562 files include: CTCF ChIP-seq (ENCFF736NYC.bed, ENCFF168IFW.bigWig), H3K27ac ChIP-seq (ENCFF544LXB.bed), RNA-PII ChIP-seq (ENCFF355MNE.bed, ENCFF914WIS.bigWig). We obtained CTCF motifs from the CTCF R package[34], which provides CTCF motifs predicted by FIMO[56]. The CTCF annotation hub we use is AH104729 (hg38.MA0139.1), and the mESC CTCF annotation hub is AH104755 (mm10.MA0139.1).

### CTCF MNase HiChIP comparison to Intact Hi-C

Publicly available K562 Intact Hi-C was downloaded from GEO (GSE237898) and compared to our MNase K562 CTCF HiChIP data. We first compared CTCF-CTCF loop calls obtained with FitHiChIP in our K562 CTCF MNase HiChIP experiment to the K562 Intact Hi-C loop calls, and found that 70% of our loop calls were also identified in the Intact Hi-C dataset (named "GSE237898_ENCFF256ZM-D_loops_GRCh38.bedpe.gz" in the GEO link GSE237898). We next examine an example individual CTCF-mediated TAD (included in both datasets' loop calls) and compare balanced contact maps for CTCF MNase HiChIP and Intact Hi-C. We observe similar structures in both assays (Supplementary Fig. 1A). To more closely examine the data at high resolution, we zoom in on the 10 kb x 10 kb window centered on the CTCF-CTCF loop marked in Supplementary Fig. 1A and plot the individual ligated fragment pairs

(Supplementary Fig. 1B). We see a strong enrichment directly at the intersection of the two CTCF sites in both datasets. While the same pattern is observed in both datasets, it is clear through the marginal density plot that chipping for CTCF in the CTCF MNase HiChIP experiment concentrates the signal at CTCF binding sites as expected (observed as a narrowing of the marginal density).

We next made a metaplot of Intact Hi-C contacts aggregated over chromosome one CTCF HiChIP loops (Supplementary Fig. 1C). We observe, as expected, a strong enrichment between loop anchors and within the enclosed chromatin loops. These analyses suggest that the overall 3D genome structure related to CTCF binding sites that we observe in CTCF MNase HiChIP is recapitulated in Intact Hi-C data.

### CAMEL nominal p-value estimation

To evaluate the significance of $\hat{\alpha}$ at a particular total read count $N = \sum_{i=1}^{4} n_i$, we simulated 100 million samples under the null hypothesis that each fragment is equally likely to occur in any of the four quadrants (Fig. 2E). This was done at each total read count ranging from 5–500. P-values at read counts beyond 500 are very similar to those at 500, so 500+ read counts are treated as bins with 500 total read count (Supplementary Fig. 8). The empirical CDF of the 100 million $\log 2(\hat{\alpha})$ at a given total read count was then computed and used to evaluate the probability of observing a value more extreme than $\log 2(\hat{\alpha})$ under the null hypothesis. The empirical CDF was evaluated at a sequence of possible $\log 2(\hat{\alpha})$ between 0 and 5 at step sizes of 0.01 (this corresponds to $\hat{\alpha} \in [1, 32]$.) This approach produces the same p-values as using $\hat{\alpha}$ instead of $\log 2(\hat{\alpha})$, but using the log enables smaller step size at large values of $\hat{\alpha}$. After acquiring the grid of p-values for each $\hat{\alpha}$ at a given read count $N$, we match the observed $\hat{\alpha}$ at a read count of $N$ with the corresponding p-value from the table. Because this approach only requires quadrant-specific read counts to match with the given table of p-values, it is very computationally efficient. Furthermore, by using the multinomial framework we place no distributional assumptions on the reads within each quadrant.

### Precision/recall curves (K562 CTCF HiChIP)

The precision recall curve is calculated for true negative and true positive CTCF binding sites in DNA loop anchors. True positives are defined as CTCF motifs that are unique within a FitHiChIP loop anchor and within 30 bp of a CTCF ChIP-seq peak. True negatives are areas of the FitHiChIP 2.5 kb loop anchors (included in the true positive set) that are at least 200 bp from the one CTCF motif. Precision is calculated as TP / (TP + FP), recall is calculated as TP / (TP + FN). TP refers to true positive, FP refers to false positive, and FN refers to false negative respectively.

### Identification of significant motifs

We use CTCF motifs identified as significant ($p < 1e\text{-}05$) by *CAMEL* as the set of CTCF binding sites. This p-value threshold was chosen based on the precision recall curve (Fig. 3B), and corresponds to the p-value at which 5% FDR is obtained.

### Short (<120 bp) fragment contact maps

We created HiChIP contact maps after filtering interactions to those involving <120 bp fragments on both sides. Filtering to the left fragment <120 bp produces a narrower peak centered on the CBS (+) (Supplementary Fig. 5B), and filtering to the right fragment <120 bp produces a narrower peak centered on the CBS (-) (Supplementary Fig. 5B).

### Comparison of replicates (K562 CTCF HiChIP)

We compared our four K562 CTCF MNase HiChIP replicates (Supplementary Fig. 3) and found a similar distribution of reads around a CBS

(Fig. 2D, Supplementary Fig. 3A). We next confirmed that the distribution of the CAMEL statistic is maintained across all four replicates at true positive CBS, defined as CTCF motifs that are unique within a loop anchor and within 30 bp of a CTCF ChIP-seq peak (Fig. 3A, Supplementary Fig. 3B). Finally, we determined that high precision is maintained across all four replicates using precision-recall curves (Fig. 3B, Supplementary Fig. 3C).

### CTCF ChIP-seq signal comparison to number of short fragments
The fraction of reads overlapping a CBS (+) that are short (<120 bp) strongly correlates with the strength of its CTCF ChIP-seq signal. CAMEL-identified CBS (+) were overlapped with CTCF ChIP-seq peaks (ENCFF736NYC.bed). CTCF ChIP-seq signalValue was cut into quartiles and CAMEL-identified CBS (+) more than 200 bp from a CTCF ChIP-seq peak were labeled as not having a CTCF ChIP-seq peak. The fraction of short reads (<120 bp) was then calculated for each CBS, plotted in a density, and colored by absence of CTCF ChIP-seq peak or CTCF ChIP-seq signalValue quartile (Fig. 4B).

### Estimating cohesin footprints
To estimate the cohesin footprints (Fig. 4C–F), we made 2D kernel density plots (bivariate normal kernel) with short (<120 bp) fragments overlapping CBS (+). Specifically, we took left fragments in an interaction pair overlapping CAMEL-identified CTCF binding sites (+) (MNase HiChIP, $n > 10,000$) or CTCF motifs within 30 bp of a CTCF ChIP-seq peak (mESC RCMC, $n = 65$). We then aligned the fragment start and fragment end relative to the start of the 35 bp CTCF motif (MA1930.1), so that "0" in the plots corresponds to the start of the 35 bp CTCF motif. (We chose this coordinate system to enable easy pinpointing of the fragment end relative to the end of the motif.) In this way we can then aggregate fragment starts and fragment ends across CBS (+) to observe the overall pattern of fragment starts and fragment ends.

The advantage of using a 2D kernel density for this aggregation is it avoids binning fragment starts and ends into arbitrary boxes (which might misrepresent the underlying signal), but instead obtains aggregated estimates of the individually-smoothed fragment starts and ends evaluated on a grid. Furthermore, density plots have the additional added advantage of automatically normalizing, which enables easy comparison of the relative prevalence of fragment start and fragment end combinations across different conditions or perturbations.

The degree of smoothing along the x (fragment start) and y (fragment end) axes is determined by the bandwidth parameter. The bandwidth parameter re-weights the distance from a specific fragment's start and end to each point (fragment start, fragment end) on the grid to determine the contribution of a fragment to the (fragment start, fragment end) point at which the 2D kernel density is being evaluated. As the bandwidth increases, the 2D kernel density estimate becomes more smoothed (fragments further away from the evaluation point contribute more.) The specific base command we are running to generate these 2D density plots (in R) is *ggplot2::stat_density_2d*, which leverages the *MASS::kde2d* function to obtain 2D kernel density estimates.

The 2D density estimation (as opposed to a 1D view) is key to obtain an accurate depiction of the fragment ends conditional on the fragment starts. Because CTCF has both a 19 bp and 35 bp motif, viewing the aggregated and smoothed fragment start, fragment end combinations is crucial to simultaneously viewing both the extended fragment end (due to cohesin) and the extended fragment start (due to the addition of the upstream motif with the core 19 bp motif).

### Estimating the fully extruded state
**K562 CTCF HiChIP.** We estimated a genome-wide range for the fully extruded state by obtaining CTCF/cohesin-protected upstream fragments (left fragment in interaction pair) overlapping CBS (+) and estimating the fraction of interaction partners overlapping a downstream convergent negative strand CTCF motif. CBS (+) were required to have at least 50 CTCF/cohesin-protected upstream fragments overlapping the motif to enable sufficient sample size for the motif-specific percent convergent calculation. We then accounted for CTCF occupancy (estimated as ~50%)[33] by dividing this estimate by two. The point estimate (5%) is the number of interaction partners overlapping a downstream convergent negative strand CTCF motif genome-wide / the total number of fragments genome-wide, and the range (1–10%) are the 1st and 99th percentile of the CBS-level CTCF-CTCF chromatin loop estimate. We confirm that these interactions are not predominantly due to random contacts by comparing to an estimate obtained using trans contacts (Supplementary Fig. 6H). Note that while some fully extruded loop interactions involve short-short fragment ligations, the majority involve one immediately CTCF-adjacent long fragment.

**mESC RCMC.** We also explored the use of RCMC for estimating fully extruded (CTCF-CTCF) loop fraction using the 150 bp mESC RCMC libraries (re-sequenced from Goel et al.[20]) covering the *Fbn2* TAD previously studied by live cell imaging in Gabriele et al.[14]. We obtain short (<120 bp) left fragments overlapping the *Fbn2* upstream TAD-anchor CBS (+) (chr18: 57,976,797–57,976,815) and estimate the frequency of interaction partners within 1kb-5kb of the CBS (-) located 505 kb downstream (chr18: 58,481,866-58,481,884) at the other end of the TAD. The estimates we obtain (1–5%) are consistent with Gabriele et al.[14] and the range reflects different window sizes: 1% of fragments are within 1 kb of the CBS (-) and 5% of fragments are within 5 kb of the CBS (-). Note that these estimates have been divided by two to adjust for the estimated ~50% occupancy of CTCF[33].

### Evaluating extruded loop size dependent on chromatin state
We used upstream fragments overlapping CAMEL-identified CTCF binding sites (+) for this analysis. 1 MB regions downstream of the CBS (+) were annotated using ChromHMM[40] to quantify the percentage of bp assigned to each of the 15 chromatin states. To simplify annotation, we grouped the 15 chromatin states into three categories (quiescent, polycomb/bivalent, and active) based on their correlation (Supplementary Fig. 6A). Regions were clustered using Ward's hierarchical clustering method[57] (Supplementary Fig. 6A). For extrusion dynamics analyses (Fig. 5C–E), each of the three chromatin categories was represented by the 20% of regions with the highest fraction of DNA in this state. Extruded loop size was then estimated as the average log10 interaction length for each annotation. Only long range TF-protected fragments (start and end at least 5 bp from motif start and end, length <120, extended fragment end (fragment end at least 48 bp from the 35 bp motif start), and interaction length >10 kb) were included in this estimate.

Similarly, high/low H3K27ac corresponds to the top 10% and bottom 10% of the number of basepairs covered by H3K27ac ChIP-seq peaks in the 1 MB regions downstream of CBS (+). High/low RNAPII corresponds to the top 10% and bottom 10% of the number of RNAPII ChIP-seq peaks located in the 1 MB regions downstream of CBS (+). Extruded loop size estimates were obtained in the same way for these annotated regions; long range TF-protected fragments with an extended fragment end were used to estimate the average log10 interaction length.

### Directionality of CBS-adjacent nucleosome position signal
Interestingly, the strength of the nucleosome positioning signal is related to the orientation of the DNA contact. Stratifying nucleosome-bound fragments based on whether they are the upstream or downstream long-range (>10 kb) fragment in a pair

(effectively single-cell left or right loop anchor) produces a differential nucleosome signal inside and outside the loop (Supplementary Fig. 9). For both upstream and downstream nucleosome-bound fragments, the nucleosome closest to the CTCF binding site and inside the loop exhibits a substantially stronger signal than the closest nucleosome outside the loop. HiChIP ligations are unlikely to fully account for this signal as a previous study using MNase-seq also showed a directional nucleosome preference around CBS (see Fig. 1A)[31].

## Reporting summary
Further information on research design is available in the Nature Portfolio Reporting Summary linked to this article.

## Data availability
The RCMC and HiChIP raw and processed data generated in this study have been deposited in the NCBI Gene Expression Omnibus (GEO) database under accession code GSE285087. Publicly available K562 ChIP-seq RAD21 BED file (Accession ID: ENCFF330SHG), CTCF BED file (Accession ID: ENCFF736NYC), CTCF bigWig signal value (Accession ID: ENCFF168IFW), RNAPII BED file (Accession ID: ENCFF355MNE), and H3K27ac BED file (Accession ID: ENCFF544LXB) were downloaded from ENCODE, and CTCF motifs were obtained from the R package *CTCF*[34] (annotation record: AH104729 (hg38.MA0139.1), documentation: https://bioconductor.org/packages/release/data/annotation/vignettes/CTCF/inst/doc/CTCF.html). K562 Intact Hi-C was downloaded from GEO (GSE237898). All K562 files downloaded were already aligned to hg38. Publicly available mESC CTCF (GSM3508478), ATAC-seq (GSE98390), H3K27ac (GSM2417096), RNAPII (GSM6809981), RAD21 (GSE137272), and SMC1A (GSE137272) were downloaded. We use mm10 coordinates for all mESC figures in this paper; files downloaded in other mouse genomes were converted to mm10 using CrossMap. CTCF motifs were obtained from the R package *CTCF*[34] (annotation record: AH104755 (mm10.MA0139.1), documentation: https://bioconductor.org/packages/release/data/annotation/vignettes/CTCF/inst/doc/CTCF.html).

## Code availability
Preprocessing, analysis and figure code used in this paper are available at https://github.com/aryeelab/cohesin_extrusion_reproducibility. Data figures in this paper were made in R v.4.1.2 using ggplot.

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

## Acknowledgements

This work was supported by the National Institutes of Health grants RM1HG009490 (MJA, JKJ, CS), R35GM118158 (JKJ), T32GM135117 (CS), and a Career Development Award from the American Society of Gene & Cell Therapy (YET). This work is supported by the Novo Nordisk Foundation NNF21SA0072102 (ASH). The content is solely the responsibility of the authors and does not necessarily represent the official views of the American Society of Gene & Cell Therapy. Dovetail Genomics / Cantata Bio supported data generation costs.

## Author contributions

C.S. and M.A. conceived the study and developed analysis methods. C.S. performed all bioinformatics analyses. E.T. and M.B. performed the HiChIP experiments. M.B., C.C., H.H., C.E., K.J., E.T., and S.J. contributed to data interpretation. V.G. and A.S.H. provided RCMC libraries and assisted in interpreting the role of cohesin in the study. M.A. supervised the overall work.

## Competing interests

Dovetail Genomics/Cantata Bio provided reagents and sample processing for HiChIP experiments. M.B. and M.S.B were employees at Dovetail Genomics during the course of this research. M.J.A has financial and consulting interests unrelated to this work in SeQure Dx and Chroma Medicine. M.J.A's interests are reviewed and managed by Dana Farber Cancer Institute. J.K.J. is a co-founder of and has a financial interest in SeQure, Dx, Inc., a company developing technologies for gene editing target profiling. JKJ also has, or had during the course of this research, financial interests in several companies developing gene editing technology: Beam Therapeutics, Blink Therapeutics, Chroma Medicine, Editas Medicine, EpiLogic Therapeutics, Excelsior Genomics, Hera Biolabs, Monitor Biotechnologies, Nvelop Therapeutics (f/k/a ETx, Inc.), Pairwise Plants, Poseida Therapeutics, and Verve Therapeutics. J.K.J.'s interests were reviewed by Massachusetts General Hospital and Mass General Brigham in accordance with their conflict of interest policies. M.J.A., Y.E.T., and J.K.J. are currently employees of Arena Bioworks. The remaining authors declare no competing interests.

## Additional information

¹Department of Biostatistics, Harvard T.H. Chan School of Public Health, Boston, MA 02115, USA. ²Department of Data Sciences, Dana-Farber Cancer Institute, Boston, MA 02115, USA. ³Broad Institute of MIT and Harvard, Cambridge, MA 02142, USA. ⁴Molecular Pathology Unit, Massachusetts General Hospital, Charlestown, MA 02129, USA. ⁵Department of Pathology, Harvard Medical School, Boston, MA 02115, USA. ⁶Department of Biological Engineering, Massachusetts Institute of Technology, Cambridge, MA 02139, USA. ⁷Koch Institute for Integrative Cancer Research, Cambridge, MA 02139, USA. ⁸The Novo Nordisk Foundation Center for Genomic Mechanisms of Disease, Broad Institute of MIT and Harvard, Cambridge, MA 02142, USA. ⁹Dovetail Genomics, Cantata Bio LLC, Scotts Valley, CA 95066, USA. ¹⁰Department of Pharmacology and Cancer Biology, Duke University School of Medicine, Durham, NC 27710, USA. ¹¹Liftoff Biosolution, Santa Cruz, CA 95060, USA. ¹²Department of Radiation Oncology, Duke University School of Medicine, Durham, NC 27710, USA. ¹³Duke Cancer Institute, Duke University School of Medicine, Durham, NC 27710, USA. ¹⁴Department of Pathology, Dana-Farber Cancer Institute, Boston, MA 02215, USA. ¹⁵Arena Bioworks, Cambridge, MA 02141, USA. ✉e-mail: martin.aryee@ds.dfci.harvard.edu

