## [Peer Review file · Nature Communications]

High-resolution CTCF footprinting reveals impact of chromatin state on cohesin extrusion

Corresponding Author: Dr Martin Aryee

Version 0:

Reviewer comments:

Reviewer #1

(Remarks to the Author)

In this manuscript Sept et al apply MNase-based HiChIP with antibodies against CTCF, and develop a novel analysis approach, taking advantage of the varying fragment sizes after MNase digestion to annotate fragments that likely originated from occupied CTCF binding sites. They also observe the footprint of cohesin next to the CTCF binding sites, discovered as a larger region of protection against MNase digestion, preferentially located downstream of the CTCF motif. And in my opinion the most exciting part of this work is estimation of extruded loop sizes using contacts where a CTCF site is present at one of the sides. This reveals that extruded loops are shorted in active regions of the genome, which authors interpret as cohesin stalling at regulatory elements. I think this is a great piece of work and a much-needed novelty in the analysis approaches for this type of data.

I don't have too many comments about the paper, but there are a few points I'd like the authors to address.

The main question for me regards the interpretation that the regulatory elements stall cohesin. This has generally been suggested in the field recently, due to contact probability patterns, especially in micro-C data. However, the data presented in this work can have alternative explanations, that I would like the authors to at least discuss, if they don't find a way to experimentally test their interpretation. For example, what immediately comes to mind is that cohesin is (potentially) preferentially loaded at regulatory elements, and then it's the higher density of cohesin and cohesin-cohesin collisions that would cause shorter loops in active regions of the genome.

Related to the above, could the authors investigate whether the direct interactions between the CTCF sites and e.g. enhancers are found more frequently than expected in their data (which would strengthen their claim of cohesin stalling), and also whether they can detect TF footprints at enhancers in their data with the same approach as they used for CTCF? I assume due to CTCF enrichment in the protocol there are much fewer reads at sites without CTCF binding sites, but I think this analysis is worth a try.

Can the authors apply their methods to published micro-C data, where there was no antibody enrichment for CTCF? If the coverage of whole-genome datasets is too low, perhaps using region-capture micro-C data would allow one to use the FactorFinder approach in an unbiased way and look for other TF footprints?

If it's possible to apply this analysis to some micro-C data from mouse ES cells, can the authors compare the frequency of the fully extruded loop at the *Fbn2* locus to the imaging data from the recent publication that they cite?

More minor points, but I would like to ask the authors to directly show an example heatmap of the data that they obtained in the figures. Perhaps it would be very interesting to see some visualization of the data that would display the effect of taking pairs with certain fragment lengths (e.g. heatmap where at least one of the interacting fragments is shorter than 120 bp, something like that).

Finally, I don't understand what figures 5c and 5d show, could the authors please clarify the legend?

Reviewer #2

(Remarks to the Author)

Cohesin and CTCF cooperate in 3D genome organization by forming and delimiting the formation of chromatin contacts, respectively. Previous studies show that only a fraction of cohesin-extruded loops connect convergent CTCF sites at any given time, while a majority of loop extrusion events are incomplete. This study combines mapping of CTCF binding and chromatin contacts in an attempt to infer the probability of 'complete' chromatin loops that connect convergent CTCF sites.

The authors perform CTCF MNase HiChIP and focus on short (<80bp) protected fragments, which - as expected - are enriched for CTCF motifs. Based on the bimodal distribution of short fragments around CTCF motifs they infer the location of CTCF binding at near base pair resolution. They call this 'FactorFinder', which seems a misnomer if the approach can find CTCF, but not other factors that don't position nucleosomes in the way CTCF does. Unless the authors can demonstrate that their method can precisely map the DNA binding sites of other factors, the name should be changed to something more CTCF-specific. While near base pair resolution mapping of CTCF binding may be of interest for some applications, it remains unclear whether it provides an advantage for identifying CTCF-based chromatin loops.

The authors find increased accessibility downstream of the CTCF motif (i.e. near the CTCF N-terminus) and ascribe this to cohesin. Although cohesin is known to accumulate at the CTCF N-terminus, the assumption that accessibility downstream of CTCF motifs is caused by cohesin needs to be tested experimentally, for example by removing cohesin and demonstrating a loss of accessibility.

The authors use their CTCF HiChIP data to estimate fully versus partially extruded CTCF-based loops. Technically, they determine the fraction of genome-wide reads that connect CTCF/cohesin-protected upstream fragments overlapping CBS+ to downstream CTCF-bound sites in convergent orientation ('full' CTCF-CTCF loops) relative to the total number of fragments genome-wide. The estimated range is referred to as 'interaction frequencies', but the methods used can only determine fractions of reads, not interaction frequencies. Although the estimates reported here are similar to previous estimates based on more direct approaches such as DNA-FISH and live cell imaging, it is unclear whether the similarity is meaningful or due to chance: There are no controls to show that reads that connect two CTCF sites or one CTCF site with non-CTCF locations actually correspond to fully or partially extruded loops, respectively. Based on the data presented, such reads may result from random ligation events or diffusive chromatin contacts, rather than cohesin-dependent extrusion. To discriminate between these possibilities, it will be necessary to eliminate cohesin-mediated loop extrusion, thus providing a background against which to measure fully or partially extruded loops.

The authors compare the length of loops formed in 'active' regions (defined by ChromHMM, H3K27ac, or RNA polymerase 2 occupancy) are longer than loops formed in inactive regions. They interpret this as an indication that active regions contain features that block cohesin-mediated loop extrusion. However, they have not excluded other interpretations, such as faster extrusion speeds or less frequent unloading of cohesin in inactive regions.

Finally, the authors propose a 'model' where transient, partially extruded states play active roles in transcription. This cannot be considered as novel because it is established that active chromatin states and genomic processes can stall cohesin-mediated loop extrusion, for example the interference of transcription with cohesin in yeast (Lengronne et al. 2004 Nature 430: 573-8) and mammalian cells (Busslinger et al., 2017 Nature 544: 503-7), boundary formation in *Drosophila* (Hou et al., 2012 Mol Cell 48: 471-84), as well as Hi-C (Bonev et al 2017 Cell 171: 557-72) and Micro-C data in mammalian cells (e.g. Hsieh et al., 2020 Mol Cell 78: 539-53). Moreover, Oh et al., 2021 Nature 595: 735-40 reported a role in transcription for 'CTCF half-loops' where CTCF binding at one loop anchor is sufficient for enhancer-promoter loops, which is essentially the model proposed here.

Reviewer #3

(Remarks to the Author)

Sept et al present a manuscript in which they describe the analysis of a CTCF MNase HiChIP dataset. Compared to regular HiChIP the MNase HiChIP (which is used in micro-C) promises a higher resolution. The authors show that short fragments (below 120bp) are enriched for CTCF binding sites, which leads the authors to conclude that these regions are protected from digestion by the binding of CTCF. Based on this an analysis method is developed, FactorFinder, that enables the identification of CTCF binding sites at high resolution. This allows them to identify loops that are supposedly mediated by CTCF. They identify loops that supposedly in the fully extruded state, which is ~1-10% of the loops. This is concordant with published numbers from live cell imaging (despite this being a completely different cell type). Their analysis further shows that loops in active chromatin regions are shorter than loops in inactive chromatin regions, from which they conclude that active chromatin impedes extrusion, a conclusion that is not necessarily warranted (see below).

In general this is a potentially interesting manuscript. However, it is not very easy to read. Also, the figure legends are very succinct (for instance Figure 5c shows correlation, correlation of what). The paper is very light on details and I do not believe that based on this information anyone would be able to reproduce this. While reviewing this paper I often had to guess how certain analyses were done and what is actually portrayed. Besides these general comments I have a number of other specific concerns.

The biggest concern is that in this paper a method is used that is not previously described (according to the references this is first paper that makes use of this method by using a commercial kit (i.e. Dovetail HiChIP)). Unfortunately, the manuscript is very light on details regarding the method. The results are described only very succinctly. The authors should compare their data to regular HiChIP and micro-C data. They should provide example plots showing contact frequency in an example

region (plots with only arcs are very difficult to interpret). In general, a much more thorough description of the data is necessary.

A second main concern is that the main conclusion of the paper (i.e. that extrusion is impeded by the chromatin state, for instance enhancers) cannot be unambiguously drawn from the analyses performed by the authors. The conclusion is drawn based on data from Figure 5. In this figure quiescent chromatin is shown to form longer loops (on average) compared to the active chromatin. Because active chromatin contains more enhancer-like chromatin (H3K27ac) and active genes (Pol II) the conclusion is drawn that these genomic elements impede extrusion. While this may be correct, the data presented here cannot be used to draw this conclusion. In quiescent chromatin there are also much fewer CTCF binding sites and there is mostly likely also less cohesin extruding through chromatin. CTCF and cohesin have been shown to block extrusion. Therefore, the lower frequency of CTCF may be a much more logical explanation why extruded loops are longer. Before the authors draw this conclusion they should take these confounding factors along in their analyses and discuss this.

Finally, with regard to the discussion of the fully extruded loop state the authors should make it clear that this is the percentage of loops that are anchored on one end by CTCF. It is likely that there are also loops that are not anchored by CTCF on both ends of the loops. This is more difficult to assess obviously, but could be very relevant for the comparison to the live cell data. A more pressing concern is how the authors envision the ligation between two distal CTCF sites occurs. For the authors to identify the fully extruded loop state two fragments of ~80bp are ligated together (if I am correct, because this is not made very clear in the Methods section). Ligation of two very small fragments seems very unlikely, where does the flexibility come from to provide ligation, when supposedly there is also at least one, but more likely two cohesin complexes present at the site of the fully extruded loop. To improve this I suggest the authors do two things:

- 1) To get a background estimation for the amount of spurious ligations between CTCF sites the authors should also quantify the number of interchromosomal interactions between CTCF sites.
- 2) Explain in the Discussion how they think the fully extruded loop state (as they envision it) can result in a ligation between genomic fragments

Other comments:

- Figure 1f: why are not all the y-axes starting at 0?
- It would be good to discuss how FactorFinder performs compared to tools such as ChIPexo and CUT&Run.
- The authors perform a simulation to calculate the precision of their method. To get a true indication of the reproducibility of the method they should perform biological replicates to determine how reproducible their loop calls are.
- Figure 4f: why is the length distribution bi-modal?
- Figure 4f: calling the fragments "TF protected" is biased (i.e. conjecture). They should just call it short and long fragments.
- Figure 5a: where does the 10kb cut-off come from. It is not rationalized anywhere in the text. I can imagine this can influence the outcome of what percentage of loops are called "fully extruded".

Version 1:

Reviewer comments:

Reviewer #1

(Remarks to the Author)

Thanks you for addressing my comments! I have nothing else to add.

(Remarks on code availability)

Reviewer #2

(Remarks to the Author)

I thank the authors for considering my concerns, which have been either addressed or acknowledged in the revised version of the manuscript.

The authors have changed the name of their method to something more appropriate and also state the potential advantages more clearly.

To address the question whether increased accessibility downstream of CTCF motifs is due to cohesin, the authors have re-sequenced region capture micro-C libraries before and after cohesin depletion. The analysis supports the claim that increased accessibility downstream of CTCF motifs is due to cohesin.

To address the question whether reads connecting CTCF sites are due to cohesin, the authors have re-sequenced region

capture micro-C libraries before and after cohesin depletion. The analysis supports the claim that reads connecting CTCF sites are largely dependent on cohesin.

The authors now explicitly acknowledge that they cannot exclude alternative explanations for the observation that loops in active regions are shorter, and describe this as a limitation of their study. As a result, i believe that the original title of the manuscript is no longer appropriate, as it is unclear whether the methods actually captures 'cohesin dynamics'.

The authors now explicitly acknowledge that their proposed model is not novel and cite relevant papers.

(Remarks on code availability)

Reviewer #3

(Remarks to the Author)

The authors have significantly improved their manuscript. I support publication of this work.

(Remarks on code availability)

RESPONSE TO REVIEWER COMMENTS FOR
High-resolution CTCF footprinting reveals impact of chromatin state on cohesin extrusion dynamics

[Author comments in blue]

[Reviewer comments in black]

We thank the reviewers for their comments which raised several important points. Through substantial revisions of the manuscript, we believe we have been able to address all the concerns raised, thereby strengthening the manuscript in multiple areas. Below, we have included a point-by-point response to the reviewer comments for your consideration.

Reviewer #1 (Remarks to the Author):

In this manuscript Sept et al apply MNase-based HiChIP with antibodies against CTCF, and develop a novel analysis approach, taking advantage of the varying fragment sizes after MNase digestion to annotate fragments that likely originated from occupied CTCF binding sites. They also observe the footprint of cohesin next to the CTCF binding sites, discovered as a larger region of protection against MNase digestion, preferentially located downstream of the CTCF motif. And in my opinion the most exciting part of this work is estimation of extruded loop sizes using contacts where a CTCF site is present at one of the sides. This reveals that extruded loops are shorted in active regions of the genome, which authors interpret as cohesin stalling at regulatory elements. I think this is a great piece of work and a much-needed novelty in the analysis approaches for this type of data.

I don't have too many comments about the paper, but there are a few points I'd like the authors to address.

- (1) The main question for me regards the interpretation that the regulatory elements stall cohesin. This has generally been suggested in the field recently, due to contact probability patterns, especially in micro-C data. However, the data presented in this work can have alternative explanations, that I would like the authors to at least discuss, if they don't find a way to experimentally test their interpretation. For example, what immediately comes to mind is that cohesin is (potentially) preferentially loaded at

regulatory elements, and then it's the higher density of cohesin and cohesin-cohesin collisions that would cause shorter loops in active regions of the genome.

We have added two additional analyses that support the claim that regulatory elements stall cohesin extrusion (see responses below - (2) and (14)). Importantly, however, (and as the reviewer points out) we are still unable to determine the mechanism by which this occurs. There is some evidence arguing against the specific hypothesis of preferential loading of cohesin at regulatory elements (e.g. [10.1073/pnas.2210480120](https://doi.org/10.1073/pnas.2210480120))¹, but it cannot be ruled out that this is relevant in certain settings. We have now included a discussion of potential mechanisms by which cohesin may be stalled at regulatory elements, but highlight that the question remains unresolved. The specific text we have added to the discussion is: "A limitation of our study, however, is that our data do not allow us to exclude alternate reasons such as differences in extrusion speed or cohesin unloading activity between active and less active chromatin regions."

- (2) Related to the above, could the authors investigate whether the direct interactions between the CTCF sites and e.g. enhancers are found more frequently than expected in their data (which would strengthen their claim of cohesin stalling), and also whether they can detect TF footprints at enhancers in their data with the same approach as they used for CTCF? I assume due to CTCF enrichment in the protocol there are much fewer reads at sites without CTCF binding sites, but I think this analysis is worth a try.

To more rigorously assess whether direct interactions between CTCF sites and enhancers are found more frequently than expected, we reanalyzed publicly available K562 intact Hi-C data (Lieberman-Aiden/ENCODE: <https://www.encodeproject.org/experiments/ENCSR479XDG/>, GEO link: <https://www.ncbi.nlm.nih.gov/geo/query/acc.cgi?acc=GSE237898>), to avoid quantification bias due to the CTCF ChIP in our HiChIP datasets.

We tested whether the direct interactions between CTCF sites and enhancers are found more frequently than expected by aggregating contacts between CTCF binding sites (CBS) and downstream enhancers, defined by H3K27ac peaks. Specifically, we used our identified CBS (+ strand) in the active group category (Supplementary Fig. 6A) and all downstream H3K27ac peaks between 10kb and 100kb away from the CBS on chromosome 1. We picked this distance range to focus on long-range interactions (>10kb) and CBS/H3K27ac pairs largely restricted to the same TAD. There are ~2700 CBS/H3K27ac pairings that match these specifications. Supplementary Fig. 6C below shows a meta-plot of detrended (observed/expected) aggregated contacts across these regions. The expected contact frequency is calculated from the distance decay $P(s)$ curve. Note that in the plot below, red corresponds to more contacts than expected (>0) and blue corresponds to less contacts than expected (<0). As expected, we observe an enrichment of contacts between the CTCF binding site (left anchor) and H3K27ac ChIP-seq

peak (right anchor). We also observe that while the CTCF/H3K27ac contact in the center of the heatmap shows the strongest interaction frequency, the stripe following the H3K27ac peak indicates that extrusion can often continue beyond the enhancer.

We also tried to apply our method to detect TF footprints at enhancers, but lacked sequencing depth to be able to use the same approach to detect other TFs without ChIP enrichment. We have renamed our method to CAMEL (CTCF Analyzer with Multinomial Estimation) to clarify that this is specifically a CTCF footprinting method.

- (3) Can the authors apply their methods to published micro-C data, where there was no antibody enrichment for CTCF? If the coverage of whole-genome datasets is too low,

perhaps using region-capture micro-C data would allow one to use the FactorFinder approach in an unbiased way and look for other TF footprints?

As the reviewer suspects, the coverage of whole genome Micro-C is insufficient for footprinting using our approach. We also tried to apply this method to RCMC data, but found that without the ChIP step enrichment for a particular transcription factor, regulatory regions are too tightly packed with multiple TFs to use this approach to identify separate footprints. We are currently working with the authors of RCMC on an alternative multi-TF footprinting strategy that takes advantage of entire fragments (rather than just the 5' end), but this work is still early.

- (4) If it's possible to apply this analysis to some micro-C data from mouse ES cells, can the authors compare the frequency of the fully extruded loop at the Fbn2 locus to the imaging data from the recent publication that they cite?

This is a great suggestion, and Reviewer 2 was similarly interested in verifying these estimates (see comment #9). To verify that we are appropriately estimating the fully and partially extruded loops, we contacted the authors of the original Region Capture Micro-C (RCMC) paper (DOI: <https://doi.org/10.1038/s41588-023-01391-1>)² and resequenced the mESC RCMC libraries covering the Fbn2 locus with 150bp paired-end reads. The libraries used in the original publication were sequenced with 50bp reads, which is too short for CTCF footprinting. Using short (<120bp) fragments that overlap the CBS (+) anchoring the left side of the Fbn2 TAD, we estimate the fraction of interaction partners within 1kb-5kb of the CBS (-) anchoring the right side of the loop as ~1-5%, consistent with Gabriele et al.'s 3-6% estimate (DOI: [10.1126/science.abn65](https://doi.org/10.1126/science.abn65))³. (The location of the CBS (+) and CBS (-), in mm10 coordinates, are chr18: 57,976,797-57,976,815 and chr18: 58,481,866-58,481,884 respectively.) This figure is now included in the supplement as Supplementary Fig. 7.

- (5) More minor points, but I would like to ask the authors to directly show an example heatmap of the data that they obtained in the figures. Perhaps it would be very interesting to see some visualization of the data that would display the effect of taking pairs with certain fragment lengths (e.g. heatmap where at least one of the interacting fragments is shorter than 120 bp, something like that).

This is a great suggestion. We have now included the example CTCF MNase HiChIP contact maps (Supplementary Fig. 5C, D) using short fragments at <120bp for consistency with the manuscript. They show contacts in a 100kb region at 1kb resolution using either all fragments (i.e. a standard contact map; left) and only those involving short fragments (<120bp; right). The short (putatively TF bound) fragment interactions highlight the interaction between two convergent CTCF binding sites (CBS) separated by ~60kb (labeled with an arrow). Note that the contact maps show log₁₀ unnormalized counts, as standard balancing is ineffective for capture assays.

To further examine the effect of restricting to short (potentially TF bound) fragments, we zoom in on the 3kb x 3kb window centered on this CBS-CBS contact and plot individual ligation pairs (Supplementary Fig. 5B below). The two CTCF binding site locations are marked with red (CBS plus strand) and blue (CBS minus strand) circles. When we subset the left fragment in the interaction pair to those shorter than 120bp, the variation along the y-axis (left fragment) is substantially reduced (top right). Similarly, when we subset the right fragment in the interaction pair to those shorter than 120bp, the variation along the x-axis (right fragment) is substantially reduced (bottom left). Filtering to both fragments shorter than 120bp produces a strong enrichment directly on the intersection of the two CBS (bottom right).

For more clarification on these plots: the y-axis is the left fragment in the interaction pair, and the x-axis is the right fragment in the interaction pair. We are graphing points corresponding to (middle of right fragment, middle of left fragment), where the middle of the fragment is defined by $(\text{fragment start} + \text{fragment end})/2$. Because of overplotting, we color the points by the number of neighbors using the `geom_pointdensity` function from the R package `ggpointdensity`. We also include the marginal densities for both left and right fragments.

(6) Finally, I don't understand what figures 5c and 5d show, could the authors please clarify the legend?

Figures 5C and 5D, now moved to the supplement, are used as justification for grouping CTCF binding site (CBS) chromatin context into three broad groupings: active, polycomb/bivalent, and quiescent. For each CBS, we quantified the fraction of 1Mb downstream chromatin classified into each of the 15 chromHMM states (i.e. X% of the 1 MB region downstream of the CBS is Quiescent, Y% is TxWk/weak transcription, etc). In Figure 5D each row corresponds to a CTCF binding site ($N \sim 10,000$), and each column shows the normalized fraction of DNA with a specific chromHMM annotation. Row clustering reveals 3 main groups. We have

removed 5C to simplify the explanation, as it was not essential. The former 5D is now Supplementary Fig. 6A.

Reviewer #2 (Remarks to the Author):

Cohesin and CTCF cooperate in 3D genome organization by forming and delimiting the formation of chromatin contacts, respectively. Previous studies show that only a fraction of cohesin-extruded loops connect convergent CTCF sites at any given time, while a majority of loop extrusion events are incomplete. This study combines mapping of CTCF binding and chromatin contacts in an attempt to infer the probability of 'complete' chromatin loops that connect convergent CTCF sites.

- (7) The authors perform CTCF MNase HiChIP and focus on short (<80bp) protected fragments, which - as expected - are enriched for CTCF motifs. Based on the bimodal distribution of short fragments around CTCF motifs they infer the location of CTCF binding at near base pair resolution. They call this 'FactorFinder', which seems a misnomer if the approach can find CTCF, but not other factors that don't position nucleosomes in the way CTCF does. Unless the authors can demonstrate that their method can precisely map the DNA binding sites of other factors, the name should be changed to something more CTCF-specific. While near base pair resolution mapping of CTCF binding may be of interest for some applications, it remains unclear whether it provides an advantage for identifying CTCF-based chromatin loops.

We have re-named our method to CAMEL (CTCF Analyzer with Multinomial Estimation) to clarify that this is specifically a CTCF footprinting method. We did not intend to suggest that this approach to footprinting CTCF binding events would be beneficial for identification of

loops. Rather we believe that the two main benefits are 1) identifying CTCF/cohesin protected fragments at the single molecule level and thereby enabling analysis of their interaction patterns, and 2) increased resolution to better distinguish between binding events at closely spaced CTCF motifs.

- (8) The authors find increased accessibility downstream of the CTCF motif (i.e. near the CTCF N-terminus) and ascribe this to cohesin. Although cohesin is known to accumulate at the CTCF N-terminus, the assumption that accessibility downstream of CTCF motifs is caused by cohesin needs to be tested experimentally, for example by removing cohesin and demonstrating a loss of accessibility.

*This is a very good suggestion, and we have sequenced and analyzed additional experimental data to address this. We turned to an mESC cohesin degron system from the Hansen lab (<https://doi.org/10.1038/s41588-023-01391-1>)² profiled with Region Capture Micro-C, in order to exclude potential biases due to the chromatin IP step in HiChIP. We re-sequenced the cohesin degron and control RCMC libraries from the original RCMC paper (<https://doi.org/10.1038/s41588-023-01391-1>)² with a longer 150bp read length to allow accurate fragment length estimation. (The original study used 50b reads.) The three loci contained in these libraries are ~1 MB each at the *Fbn2*, *Ppm1g*, and *Klf1* genes. To visualize fragments overlapping CTCF motifs, we use 2D density plots where each data point represents a fragment's start (x-axis) and end (y-axis) position relative to the start of the 35bp JASPAR CTCF motif (MA1930.1). Fragment start and fragment end have been aligned such that 0 represents the start of the 35bp motif. An aggregate plot of short (<120bp) fragments overlapping 65 bound CTCF motifs on the positive strand (defined as a CTCF motif (+) within 30bp of a CTCF ChIP-seq peak) in the DMSO control condition shows an enrichment of fragments that extend ~15bp downstream of the CTCF motif footprint (red box). In the cohesin depletion condition, these 'downstream extended footprint' fragments are depleted, and the most prevalent fragment size now spans only the CTCF motif itself (blue box).*

In summary, we observe an extended footprint downstream of CTCF binding sites using an independent assay (RCMC vs HiChIP), and in a different cell line (mESC compared to K562), and this downstream footprint is cohesin-dependent. This figure is now included as Figure 4E, F.

mESC RCMC

Most Prevalent Fragment Type

- (9) The authors use their CTCF HiChIP data to estimate fully versus partially extruded CTCF-based loops. Technically, they determine the fraction of genome-wide reads that connect CTCF/'cohesin'-protected upstream fragments overlapping CBS+ to downstream CTCF-bound sites in convergent orientation ('full' CTCF-CTCF loops) relative to the total number of fragments genome-wide. The estimated range is referred to as 'interaction frequencies', but the methods used can only determine fractions of reads, not interaction frequencies. Although the estimates reported here are similar to previous estimates based on more direct approaches such as DNA-FISH and live cell imaging, it is unclear whether the similarity is meaningful or due to chance: There are no controls to show that reads that connect two CTCF sites or one CTCF site with non-CTCF locations actually correspond to fully or partially extruded loops, respectively. Based on the data presented, such reads may result from random ligation events or diffusive chromatin contacts, rather than cohesin-dependent extrusion. To discriminate between these possibilities, it will be necessary to eliminate cohesin-mediated loop extrusion, thus providing a background against which to measure fully or partially extruded loops.

To verify that 'fully extruded' loop estimates represent cohesin-dependent interactions, we obtained Region Capture Micro-C (RCMC) data from the Fbn2 locus in a mESC cohesin degron cell line. We resequenced the WT, RAD21 degron and DMSO control libraries used in the original RCMC paper (DOI: <https://doi.org/10.1038/s41588-023-01391-1>)² with 150bp paired-end reads, as the published 50bp reads are too short for CTCF footprinting. From the WT condition, we estimate that the TAD spends ~1-5% of its lifetime fully extruded, in agreement with the ~3-6% estimate obtained by live cell imaging of this locus (DOI: [10.1126/science.abn65](https://doi.org/10.1126/science.abn65))³. To evaluate the relative contributions of cohesin extrusion vs diffusive chromatin/random ligation contacts we used a pseudo-4C analysis with a viewpoint at the CTCF binding site on one end of the Fbn2 TAD. Cohesin depletion (IAA) reduces the 'fully extruded' contacts with the convergent CBS at the other end of the TAD by ~75% compared to

the DMSO control condition. (See red box in Supplementary Fig. 7B below).

- (10) The authors compare the length of loops formed in 'active' regions (defined by ChromHMM, H3K27ac, or RNA polymerase 2 occupancy) are longer than loops formed in inactive regions. They interpret this as an indication that active regions contain features that block cohesin-mediated loop extrusion. However, they have not excluded other interpretations, such as faster extrusion speeds or less frequent unloading of cohesin in inactive regions.

We agree with this important point. To partially address this, we have added additional analyses showing increased contact frequency between CTCF sites and downstream enhancers and TSSs (see Supplementary Fig. 6C), suggesting that these features may be part of the explanation for differences in loop lengths. Nonetheless, our data do not allow us to exclude reasons such as differences in extrusion speed or cohesin unloading activity between active and less active chromatin regions, and we have added this limitation to the discussion to make this limitation clear. The specific text we have added to the discussion is: “A limitation of our study, however, is that our data do not allow us to exclude alternate reasons such as differences in extrusion speed or cohesin unloading activity between active and less active chromatin regions.”

- (11) Finally, the authors propose a 'model' where transient, partially extruded states play active roles in transcription. This cannot be considered as novel because it is established that active chromatin states and genomic processes can stall cohesin-mediated loop extrusion, for example the interference of transcription with cohesin in yeast (Lengronne et al. 2004 Nature 430: 573-8) and mammalian cells (Busslinger et al., 2017 Nature 544: 503-7), boundary formation in Drosophila (Hou et al., 2012 Mol Cell 48: 471-84), as well as Hi-C (Bonev et al 2017 Cell 171: 557-72) and Micro-C data in mammalian cells (e.g. Hsieh et al., 2020 Mol Cell 78: 539-53). Moreover, Oh et al., 2021 Nature 595: 735-40 reported a role in transcription for 'CTCF half-loops' where CTCF binding at one loop anchor is sufficient for enhancer-promoter loops, which is essentially the model proposed here.

This point is well noted. We have updated the text to clarify that our estimates about partially extruded loop frequency and size in different chromatin contexts are consistent with the existing work cited above, and have included the citations in the revised manuscript.

Reviewer #3 (Remarks to the Author):

Sept et al present a manuscript in which they describe the analysis of a CTCF MNase HiChIP dataset. Compared to regular HiChIP the MNase HiChIP (which is used in micro-C) promises a higher resolution. The authors show that short fragments (below 120bp) are enriched for CTCF binding sites, which leads the authors to conclude that these regions are protected from digestion by the binding of CTCF. Based on this an analysis method is developed, FactorFinder, that enables the identification of CTCF binding sites at high resolution. This allows them to identify loops that are supposedly mediated by CTCF. They identify loops that supposedly in the fully extruded state, which is ~1-10% of the loops. This is concordant with published numbers from live cell imaging (despite this being a completely different cell type).

Their analysis further shows that loops in active chromatin regions are shorter than loops in inactive chromatin regions, from which they conclude that active chromatin impedes extrusion, a conclusion that is not necessarily warranted (see below).

- (12) In general this is a potentially interesting manuscript. However, it is not very easy to read. Also, the figure legends are very succinct (for instance Figure 5c shows correlation, correlation of what). The paper is very light on details and I do not believe that based on this information anyone would be able to reproduce this. While reviewing this paper I often had to guess how certain analyses were done and what is actually portrayed. Besides these general comments I have a number of other specific concerns.

We agree that reproducibility and transparency in how results were obtained is crucial, and with this in mind we created a public code repository with all the necessary code needed to reproduce every figure and result in our paper. This is mentioned in our paper in the “Code availability” section, and we have also provided the link to our publicly available github site here for reference (https://github.com/aryeelab/cohesin_extrusion_reproducibility). Both human readable .md files and code .Rmd files are provided for both data processing (Protocol.Rmd) and each figure (labeled Figure1.Rmd, Figure2.Rmd, etc.).

We have also expanded and clarified our methods descriptions and figure legends throughout the manuscript (including for Figure 5 as noted above).

- (13) The biggest concern is that in this paper a method is used that is not previously described (according to the references this is first paper that makes use of this method by using a commercial kit (i.e. Dovetail HiChIP)). Unfortunately, the manuscript is very light on details regarding the method. The results are described only very succinctly. The authors should compare their data to regular HiChIP and micro-C data. They should provide example plots showing contact frequency in an example region (plots with only arcs are very difficult to interpret). In general, a much more thorough description of the data is necessary.

This is a good point, and we should have made more clear in the reference list that this is not the first paper to use the Dovetail MNase HiChIP kit. We now cite references for a set of papers that have previously used this kit (<https://doi.org/10.1016/j.cell.2022.12.027>, <https://doi.org/10.1038/s41586-021-04246-z>, [10.1093/nar/gkad361](https://doi.org/10.1093/nar/gkad361))⁴⁻⁶.

We have also included comparative analyses of our Dovetail HiChIP data with public ENCODE MNase Hi-C (Intact Hi-C) data. (ENCODE: <https://www.encodeproject.org/experiments/ENCSR479XDG/>, GEO:

<https://www.ncbi.nlm.nih.gov/geo/query/acc.cgi?acc=GSE237898>). We first compared CTCF-CTCF loop calls obtained with FitHiChIP in our K562 CTCF MNase HiChIP experiment to the K562 Intact Hi-C loop calls, and found that 70% of our loop calls were also identified in Erez Aiden's dataset (named "GSE237898_ENCFF256ZMD_loops_GRCh38.bedpe.gz" in the GEO link previously shown).

We next examine an example individual CTCF-mediated TAD (included in both datasets' loop calls) and compare balanced contact maps (2kb resolution) for CTCF MNase HiChIP (above the diagonal) and Intact HiC (below the diagonal), and observe similar structures in both assays. This figure is included as Supplementary Fig. 1A in the paper.

To more closely examine the data at high resolution, we zoom in on the 10kb x 10kb window centered on the CTCF-CTCF loop marked by the arrow above and plot the individual ligated fragment pairs. We see a strong enrichment directly at the intersection of the two CTCF sites in both datasets. Note that we have colored the points by the number of neighbors, since they are overplotted. We have additionally included marginal densities for the left and right fragments in the interaction pair (5' end of the read). While the same pattern is observed in both datasets, it is clear through the marginal density plot that chipping for CTCF in the CTCF MNase HiChIP experiment (top figure) concentrates the signal at CTCF binding sites as expected (observed as a narrowing of the marginal density). This figure is included as Supplementary Fig. 1B in the paper.

B
We next made a metaplot of Intact Hi-C contacts aggregated over chromosome one CTCF HiChIP loops ($n = 3645$ loops, resolution = 4kb, linear scale, detrended contacts. Red represents more contacts than expected while blue represents less contacts than expected). Aggregated contacts are obtained using the `HiContacts::aggregate` R function. We observe, as expected, a strong enrichment between loop anchors and within the enclosed chromatin loops (see plot below). This figure is included as Supplementary Fig. 1C in the paper.

These analyses suggest that the overall 3D genome structure related to CTCF binding sites that we observe in CTCF HiChIP is recapitulated in Intact Hi-C data. These analyses have now been included in the supplemental material and are referenced from the main text.

(14) A second main concern is that the main conclusion of the paper (i.e. that extrusion is impeded by the chromatin state, for instance enhancers) cannot be unambiguously drawn from the analyses performed by the authors. The conclusion is drawn based on data from Figure 5. In this figure quiescent chromatin is shown to form longer loops (on average) compared to the active chromatin. Because active chromatin contains more enhancer-like chromatin (H3K27ac) and active genes (Pol II) the conclusion is drawn that these genomic elements impede extrusion. While this may be correct, the data presented here cannot be used to draw this conclusion. In quiescent chromatin there are also much fewer CTCF binding sites and there is mostly likely also less cohesin extruding through chromatin. CTCF and cohesin have been shown to block extrusion. Therefore, the lower frequency of CTCF may be a much more logical explanation why extruded loops are longer. Before the authors draw this conclusion they should take these confounding factors along in their analyses and discuss this.

We agree that CTCF is a very important confounding factor to consider in this analysis, and have included two analyses to address this:

1) We repeated the loop length analysis using short (putatively CTCF/cohesin bound) fragments overlapping positive strand CTCF binding sites, "CBS(+)", but removed those with partner fragments within 1kb of a CTCF motif on the negative strand (thus removing CTCF-CTCF contacts). This analysis confirms that the effect of chromatin state on loop length remains after removing contacts between convergent CTCF motifs (Supplementary Fig. 6E, F below).

2) We also performed a high-resolution locus-specific analysis using Region Capture Micro-C data and asked if short fragments preferentially contact regulatory elements without CTCF binding sites, and whether these contacts are cohesin-dependent. We resequenced the mESC WT, RAD21 degon and DMSO control libraries covering the mESC Fbn2 locus used in the original RCMC paper (DOI: <https://doi.org/10.1038/s41588-023-01391-1>)² with 150bp paired-end reads, as the published 50bp reads are too short for CTCF footprinting. We make a pseudo-4C plot, by filtering to CTCF-bound fragments for the left anchor of the Fbn2 TAD (the

viewpoint), and plot the location of the interaction partner of each fragment in the DMSO control and cohesin depletion conditions. We find an enrichment of contacts between CTCF-bound fragments at the left TAD anchor and the H3K27Ac and RNAPII peaks at the TSS of *Fbn2*, and observe that this enrichment is visibly attenuated upon cohesin depletion (Supplementary Fig. 7B below). Note that there are also ChIP-Seq peaks for RNAPII and the cohesin subunits RAD21 and SMC1A at the TSS of *Fbn2*. The co-occurrence of RNAPII ChIP-seq peaks with cohesin subunit ChIP-seq peaks in the absence of CTCF has been previously observed in mESC (DOI: [10.1101/gr.136507.111](https://doi.org/10.1101/gr.136507.111))⁷. In summary, the cohesin-dependent enrichment of Region Capture Micro-C contacts between the viewpoint CBS and the downstream H3K27ac & RNAPII-marked regulatory region without a CBS, further suggests that presence of regulatory elements can shorten loops independent of CTCF binding sites.

3D interaction patterns at the mESC *Fbn2* locus. (**Top panel**) RCMC-derived 1D density estimation for the interaction partners of CTCF-bound fragments at the marked viewpoint. (**Bottom panel**) Protein binding (ChIP-seq), enhancer activity (H3K27ac) and open chromatin (ATAC-seq) annotation from ENCODE.

- (15) Finally, with regard to the discussion of the fully extruded loop state the authors should make it clear that this is the percentage of loops that are anchored on one end by CTCF. It is likely that there are also loops that are not anchored by CTCF on both ends

of the loops. This is more difficult to assess obviously, but could be very relevant for the comparison to the live cell data.

We have modified the manuscript to make this point clear. This is now included in the Results section titled “Regulatory elements obstruct cohesin-mediated loop extrusion” in the sentence “We note that these fully extruded loop estimates are contingent on at least one end of a long-range interaction being anchored by CTCF, and does not include long-range interactions that are not anchored by CTCF on either end (the fully unlooped state).”

(16) A more pressing concern is how the author envision the ligation between two distal CTCF sites occurs. For the authors to identify the fully extruded loop state two fragments of ~80bp are ligated together (if I am correct, because this is not made very clear in the Methods section). Ligation of two very small fragments seems very unlikely, where does the flexibility come from to provide ligation, when supposedly there is also at least one, but more likely two cohesin complexes present at the site of the fully extruded loop. To improve this I suggest the authors do two things:

1) To get a background estimation for the amount of spurious ligations between CTCF sites the authors should also quantify the number of interchromosomal interactions between CTCF sites.

We agree this is an important control. For each CTCF binding site covered by at least 50 fragments, we computed the fraction of intra- and inter-chromosomal contacts where the partner overlaps another CTCF motif. The median fraction of CTCF-CTCF contacts using inter-chromosomal pairs, likely representing the background signal level, is 1%, compared to 5% for intra-chromosomal pairs. This figure is now included as Supplementary Fig. 6H.

2) Explain in the Discussion how they think the fully extruded loop state (as they envision it) can result in a ligation between genomic fragments

This comment raises an important point where we should have been more clear in the original manuscript. While some fully extruded loop interactions do involve short-short fragment ligations, the majority involve one immediately CTCF-adjacent long fragment. We have updated the manuscript discussion to clarify this point (see the “Estimating the fully extruded state” section of Methods.)

Other comments:

(17) Figure 1f: why are not all the y-axes starting at 0?

Thanks for catching this! We’ve updated this figure with all the y-axes starting at 0. I’ve inserted the figure here for reference.

(18) It would be good to discuss how FactorFinder performs compared to tools such as ChIPexo and CUT&Run.

We don't expect our approach would outperform ChIP-exo or CUT&Run, but rather believe the strength is that we can achieve similar resolution using proximity ligation data and also integrate this with chromatin interaction information. We have added these points to the Discussion (see the second paragraph).

- (19) The authors perform a simulation to calculate the precision of their method. To get a true indication of the reproducibility of the method they should perform biological replicates to determine how reproducible their loop calls are.

This is a great suggestion. The figures we have in the paper are pooling together four biological replicates, so we were able to repeat the analyses with each of the four replicates individually. We have included additional plots (shown below), separating data by replicate.

In figure 2D, we show a density plot of the fragments at an example CBS on chr1 (top plot below). If we make this same plot by replicate (bottom plot below), we can see this observed signal is concordant across replicates.

We also plot individual replicate versions of Figure 3A (which used all 4 replicates) to evaluate the reproducibility of footprinting CTCF binding events. The Figures show CTCF motifs that are unique within a loop anchor and within 30 bp of a CTCF ChIP-seq peak.

Finally, we examined precision-recall curves by replicate. At a p-value threshold of $1e-03$, we achieve a precision of $> 90\%$ across all four replicates (points labeled with “+”). However,

likely due to the decrease in the number of reads used, the recall is decreased in the replicate-by-replicate analysis. These replicate plots are included in the supplement as Supplementary Fig. 3A-C.

(20) Figure 4f: why is the length distribution bi-modal?

This is largely a consequence of the log10 scale used on the x-axis, which assigns increasingly large distance bins to the same x-axis step. (e.g. both 10kb-100kb and 100kb - 1MB represent one x-axis unit). This effectively pushes the long-range interactions together, forming a second peak.

(21) Figure 4f: calling the fragments “TF protected” is biased (i.e. conjecture). They should just call it short and long fragments.

We have taken your suggestion and re-labeled Figures 4H and 4I to “short fragments”, “short fragments, extended end”, and “long fragments”. We also have added a schematic (Fig. 4G) to help readers interpret Fig. 4H and 4I, but have made it distinct from Fig. 4H and Fig. 4I so readers can distinguish between the data and interpretation.

G**H****I**
- (22) Figure 5a: where does the 10kb cut-off come from. It is not rationalized anywhere in the text. I can imagine this can influence the outcome of what percentage of loops are called “fully extruded”.

10kb was chosen based on the $P(s)$ curve shown below (Supplementary Fig. 4A). It shows similar behavior for interactions in the <10kb range suggesting that random polymer interactions dominate at these length scales. In contrast, interactions involving smaller protein (possibly TF) footprints are enriched in the >10kb distance range. Therefore, filtering to interactions > 10kb was chosen to exclude interactions in the range where cohesin extrusion is less relevant, consistent with⁸⁻¹⁰.

A
REFERENCES

1. Banigan, E. J. *et al.* Transcription shapes 3D chromatin organization by interacting with loop extrusion. *Proc. Natl. Acad. Sci.* **120**, e2210480120 (2023).
2. Goel, V. Y., Huseyin, M. K. & Hansen, A. S. Region Capture Micro-C reveals coalescence of enhancers and promoters into nested microcompartments. *Nat. Genet.* 1–9 (2023) doi:10.1038/s41588-023-01391-1.
3. Gabriele, M. *et al.* Dynamics of CTCF- and cohesin-mediated chromatin looping revealed by live-cell imaging. *Science* **376**, 496–501 (2022).
4. Yang, J.-H. *et al.* Loss of epigenetic information as a cause of mammalian aging. *Cell* **186**, 305–326.e27 (2023).
5. Xiao, L. *et al.* Targeting SWI/SNF ATPases in enhancer-addicted prostate cancer. *Nature* **601**, 434–439 (2022).
6. Kim, H.-M. *et al.* Forkhead box protein D2 suppresses colorectal cancer by reprogramming enhancer interactions. *Nucleic Acids Res.* **51**, 6143–6155 (2023).
7. Faure, A. J. *et al.* Cohesin regulates tissue-specific expression by stabilizing highly occupied

- cis-regulatory modules. *Genome Res.* **22**, 2163–2175 (2012).
8. Bhattacharyya, S., Chandra, V., Vijayanand, P. & Ay, F. Identification of significant chromatin contacts from HiChIP data by FitHiChIP. *Nat. Commun.* **10**, 4221 (2019).
 9. Roayaei Ardakany, A., Gezer, H. T., Lonardi, S. & Ay, F. Mustache: multi-scale detection of chromatin loops from Hi-C and Micro-C maps using scale-space representation. *Genome Biol.* **21**, 256 (2020).
 10. Lareau, C. A. & Aryee, M. J. hichipper: a preprocessing pipeline for calling DNA loops from HiChIP data. *Nat. Methods* **15**, 155–156 (2018).

RESPONSE TO REVIEWER COMMENTS FOR
High-resolution CTCF footprinting reveals impact of chromatin state on cohesin extrusion

[Author comments in blue]

[Reviewer comments in black]

We thank the reviewers for their comments and feedback. We have followed the suggestion of Reviewer 2 and updated the manuscript title (removing the word dynamics) to address their concerns. Below, we have included a point-by-point response to the reviewer comments for your consideration.

REVIEWERS' COMMENTS

Reviewer #1 (Remarks to the Author):

Thanks you for addressing my comments! I have nothing else to add.

Thank you for your insightful feedback, addressing your comments has significantly improved the manuscript!

Reviewer #2 (Remarks to the Author):

I thank the authors for considering my concerns, which have been either addressed or acknowledged in the revised version of the manuscript.

The authors have changed the name of their method to something more appropriate and also state the potential advantages more clearly.

To address the question whether increased accessibility downstream of CTCF motifs is due to cohesin, the authors have re-sequenced region capture micro-C libraries before and after cohesin depletion. The analysis supports the claim that increased accessibility downstream of CTCF motifs is due to cohesin.

To address the question whether reads connecting CTCF sites are due to cohesin, the authors have re-sequenced region capture micro-C libraries before and after cohesin depletion. The analysis supports the claim that reads connecting CTCF sites are largely dependent on cohesin.

The authors now explicitly acknowledge that they cannot exclude alternative explanations for

the observation that loops in active regions are shorter, and describe this as a limitation of their study. As a result, i believe that the original title of the manuscript is no longer appropriate, as it is unclear whether the methods actually captures 'cohesin dynamics'.

The authors now explicitly acknowledge that their proposed model is not novel and cite relevant papers.

Thank you for your insightful feedback, addressing your comments has significantly improved the manuscript! We have updated our manuscript title to “High-resolution CTCF footprinting reveals impact of chromatin state on cohesin extrusion”, removing the word “dynamics” from the title to address your concern that it is unclear whether we are truly capturing cohesin extrusion dynamics.

Reviewer #3 (Remarks to the Author):

The authors have significantly improved their manuscript. I support publication of this work.

Thank you for your insightful feedback, addressing your comments has significantly improved the manuscript!